# Self-Consistent Velocity Matching of Probability Flows

**Lingxiao Li**
MIT CSAIL
lingxiao@mit.edu

**Samuel Hurault**
Univ. Bordeaux, Bordeaux INP, CNRS, IMB
samuel.hurault@math.u-bordeaux.fr

**Justin Solomon**
MIT CSAIL
jsolomon@mit.edu

## Abstract

We present a discretization-free scalable framework for solving a large class of mass-conserving partial differential equations (PDEs), including the time-dependent Fokker-Planck equation and the Wasserstein gradient flow. The main observation is that the time-varying velocity field of the PDE solution needs to be *self-consistent*: it must satisfy a fixed-point equation involving the probability flow characterized by the same velocity field. Instead of directly minimizing the residual of the fixed-point equation with neural parameterization, we use an iterative formulation with a biased gradient estimator that bypasses significant computational obstacles with strong empirical performance. Compared to existing approaches, our method does not suffer from temporal or spatial discretization, covers a wider range of PDEs, and scales to high dimensions. Experimentally, our method recovers analytical solutions accurately when they are available and achieves superior performance in high dimensions with less training time compared to alternatives.

## 1 Introduction

Mass conservation is a ubiquitous phenomenon in dynamical systems arising from fluid dynamics, electromagnetism, thermodynamics, and stochastic processes. Mathematically, mass conservation is formulated as the *continuity equation*:

$$\partial_t p_t(x) = -\boldsymbol{\nabla} \cdot (v_t p_t), \forall x, t \in [0, T], \tag{1}$$

where $p_t : \mathbf{R}^d \to \mathbf{R}$ is a scalar quantity such that the total mass $\int p_t(x)$ is conserved with respect to $t$, $v_t : \mathbf{R}^d \to \mathbf{R}^d$ is a velocity field, and $T > 0$ is total time. We will assume, for all $t \in [0, T]$, $p_t \geq 0$ and $\int p_t(x) \, \mathrm{d}x = 1$, i.e., $p_t$ is a probability density function. We use $\mu_t$ to denote the probability measure with density $p_t$. Once a pair $(p_t, v_t)$ satisfies (1), the density $p_t$ is coupled with $v_t$ in the sense that the evolution of $p_t$ in time is characterized by $v_t$ (Section 3.1).

We consider the subclass of mass-conserving PDEs that can be written succinctly as

$$\partial_t p_t(x) = -\boldsymbol{\nabla} \cdot (f_t(x; \mu_t) p_t), \forall x, t \in [0, T], \tag{2}$$

where $f_t(\cdot; \mu_t) : \mathbf{R}^d \to \mathbf{R}^d$ is a given function depending on $\mu_t$, with initial condition $\mu_0 = \mu_0^*$ for a given initial probability measure $\mu_0^*$ with density $p_0^*$.

Different choices of $f_t$ lead to a large class of mass-conserving PDEs. For instance, given a functional $\mathcal{F} : \mathcal{P}_2(\mathbf{R}^d) \to \mathbf{R}$ on the space of probability distributions with finite second moments, if we take

$$f_t(x; \mu_t) := -\nabla_{W_2} \mathcal{F}(\mu_t)(x), \tag{3}$$

where $\nabla_{W_2} \mathcal{F}(\mu) : \mathbf{R}^d \to \mathbf{R}^d$ is the Wasserstein gradient of $\mathcal{F}$, then the solution to (2) is the Wasserstein gradient flow of $\mathcal{F}$ [Santambrogio, 2015, Chapter 8]. Thus, solving (2) efficiently allows us to optimize in the probability measure space. If we take

$$f_t(x; \mu_t) := b_t(x) - D_t(x)\nabla \log p_t(x), \tag{4}$$

where $b_t$ is a velocity field and $D_t(x)$ is a positive-semidefinite matrix, then we obtain the time-dependent Fokker-Planck equation [Risken and Risken, 1996], which describes the time evolution of the probability flow undergoing drift $b_t$ and diffusion with coefficient $D_t$.

A popular strategy to solve (2) is to use an Eulerian representation of the density field $p_t$ on a discretized mesh or as a neural network [Raissi et al., 2019]. However, these approaches do not fully exploit the mass-conservation principle and are usually limited to low dimensions. Shen et al. [2022], Shen and Wang [2023] recently introduced the notion of *self-consistency* for the Fokker-Planck equation and more generally McKean-Vlasov type PDEs. This notion is a Lagrangian formulation of (2). They apply the adjoint method to optimize self-consistency. In this work, we extend their notion of self-consistency to mass-conserving PDEs of the general form (2). Equipped with this formulation, we develop an iterative optimization scheme called *self-consistent velocity matching*. With the probability flow parameterized as a neural network, at each iteration, we refine the velocity field $v_t$ of the current flow to match an estimate of $f_t$ evaluated using the network weights from the previous iteration. Effectively, we minimize the self-consistency loss with a biased but more tractable gradient estimator.

This simple scheme has many benefits. First, the algorithm is agnostic to the form of $f_t$, thus covering a wider range of PDEs compared to past methods. Second, we no longer need to differentiate through differential equations using the adjoint method as in Shen and Wang [2023], which is orders of magnitude slower than our method with worse performance in high dimensions. Third, this iterative formulation allows us to rewrite the velocity-matching objectives for certain PDEs to get rid of computationally expensive quantities such as $\nabla \log p_t$ in the Fokker-Planck equation (Proposition 3.1). Lastly, our method is flexible with probability flow parameterization: we have empirically found that the two popular ways of parameterizing the flow—as a time-varying pushforward map [Biloš et al., 2021] and as a time-varying velocity field [Chen et al., 2018]—both have merits in different scenarios.

Our method tackles mass-conserving PDEs of the form (2) in a unified manner without temporal or spatial discretization. Despite using a biased gradient estimator, in practice, our method decreases the self-consistency loss efficiently (second column of Figure 2). For PDEs with analytically-known solutions, we quantitatively compare with the recent neural JKO-based methods [Mokrov et al., 2021, Fan et al., 2021, Alvarez-Melis et al., 2021], the adjoint method [Shen and Wang, 2023], and the particle-based method [Boffi and Vanden-Eijnden, 2023]. Our method faithfully recovers true solutions with quality on par with the best previous methods in low dimensions and with superior quality in high dimensions. Our method is also significantly faster than competing methods, especially in high dimensions, at the same time without discretization. We further demonstrate the flexibility of our method on two challenging experiments for modeling flows splashing against obstacles and smooth interpolation of measures where the comparing methods are either not applicable or have noticeable artifacts.

## 2 Related Works

Classical PDE solvers for mass-conserving PDEs such as the Fokker-Planck equation and the Wasserstein gradient flow either use an Eulerian representation of the density and discretize space as a grid or mesh [Burger et al., 2010, Carrillo et al., 2015, Peyré, 2015] or use a Lagrangian representation, which discretizes the flow as a collection of interacting particles simulated forward in time [Crisan and Lyons, 1999, Westdickenberg and Wilkening, 2010]. Due to spatial discretization, these methods struggle with high-dimensional problems. Hence, the rest of the section focuses solely on recent neural network-based methods.

**Physics-informed neural networks.** Physics-informed neural networks (PINNs) are prominent methods that solve PDEs using deep learning [Raissi et al., 2019, Karniadakis et al., 2021]. The main idea is to minimize the residual of the PDE along with loss terms to enforce the boundary conditions and to match observed data. Our notion of self-consistency is a Lagrangian analog of the residual in PINN. Our velocity matching only occurs along the flow of the current solution where interesting dynamics happen, while in PINNs the residual is evaluated on collocation points that occupy the entire domain. Hence our method is particularly suitable for high-dimensional problems where the dynamics have a low-dimensional structure.

**Neural JKO methods.** Recent works [Mokrov et al., 2021, Alvarez-Melis et al., 2021, Fan et al., 2021] apply deep learning to the time-discretized JKO scheme [Jordan et al., 1998] to solve Wasserstein gradient flow (3). By pushing a reference measure through a chain of neural networks parameterized as input-convex neural networks (ICNNs) [Amos et al., 2017], these methods avoid discretizing the space. Mokrov et al. [2021] optimize one ICNN to minimize Kullback-Leibler (KL) divergence plus a Wasserstein-2 distance term at each JKO step. This method is extended to other functionals by Alvarez-Melis et al. [2021]. Fan et al. [2021] use the variational formulation of $f$-divergence to obtain a faster primal-dual approach.

An often overlooked problem of neural JKO methods is that the total training time scales quadratically with the number of JKO steps: to draw samples for the current step, initial samples from the reference measure must be passed through a long chain of neural networks, along with expensive quantities like densities. However, using too few JKO steps results in large temporal discretization errors. Moreover, the optimization at each step might not have fully converged before the next step begins, resulting in an unpredictable accumulation of errors. In contrast, our method does not suffer from temporal discretization and can be trained end-to-end. It outperforms these neural JKO methods with less training time in experiments we considered.

**Velocity matching.** A few recent papers employ the idea of velocity matching to construct a flow that follows a learned velocity field. di Langosco et al. [2021] simulate the Wasserstein gradient flow of the KL divergence by learning a velocity field that drives a set of particles forward in time for Bayesian posterior inference. The velocity field is refined on the fly based on the current positions of the particles. Boffi and Vanden-Eijnden [2023] propose a similar method that applies to a more general class of time-dependent Fokker-Planck equations. These two methods approximate probability measures using finite particles which might not capture high-dimensional distributions well. Liu et al. [2022], Lipman et al. [2022], Albergo and Vanden-Eijnden [2022] use flow matching for generative modeling by learning a velocity field that generates a probability path connecting a reference distribution to the data distribution. Yet these methods are not designed for solving PDEs.

Most relevant to our work, Shen et al. [2022] propose the concept of self-consistency for the Fokker-Planck equation, later extended to McKean-Vlasov type PDEs [Shen and Wang, 2023]. They observe that the velocity field of the flow solution to the Fokker-Planck equation must satisfy a fixed-point equation. They theoretically show that, under certain regularity conditions, a form of probability divergence between the current solution and the true solution is bounded by the self-consistency loss that measures the violation of the fixed-point equation. Their algorithm minimizes such violation using neural ODE parameterization [Chen et al., 2018] and the adjoint method. Our work extends the concept of self-consistency to a wider class of PDEs in the form of (2) and circumvents the computationally demanding adjoint method using an iterative formulation. We empirically verify that our method is significantly faster and reduces the self-consistency loss more effectively in moderate dimensions than that of Shen and Wang [2023] (Figure 2).

## 3   Self-Consistent Velocity Matching

### 3.1   Probability flow of the continuity equation

A key property of the continuity equation (1) is that any solution $(p_t, v_t)_{t \in [0,T]}$ (provided $p_t$ is continuous with respect to $t$ and $v_t$ is bounded) corresponds to a unique flow map $\{\Phi_t(\cdot) : \mathbf{R}^d \to \mathbf{R}^d\}_{t \in [0,T]}$ that solves the ordinary differential equations (ODEs) [Ambrosio et al., 2005, Proposition 8.1.8]

$$\Phi_0(x) = x, \frac{\mathrm{d}}{\mathrm{d}t}\Phi_t(x) = v_t(\Phi_t(x)), \forall x, t \in [0, T], \tag{5}$$

and the flow map satisfies $\mu_t = (\Phi_t)_\# \mu_0$ for all $t \in [0, T]$, where $(\Phi_t)_\# \mu_0$ to denote the pushforward measure of $\mu_0$ by $\Phi_t$. Moreover, the converse is true: any solution $(\Phi_t, v_t)$ of (5) with Lipschitz continuous and bounded $v_t$ is a solution of (1) with $\mu_t = (\Phi_t)_\# \mu_0$ [Ambrosio et al., 2005, Lemma 8.1.6]. Thus the Eulerian viewpoint of (1) is equivalent to the Lagrangian viewpoint of (5). We next exploit this equivalence by modeling the probability flow using the Lagrangian viewpoint so that it automatically satisfies the continuity equation (1).

## 3.2 Parametrizing probability flows

Our algorithm will be agnostic to the exact parameterization used to represent the probability flow. As such, we need a way to parameterize the flow to access the following quantities for all $t \in [0, T]$:

- $\Phi_t : \mathbf{R}^d \to \mathbf{R}^d$, the flow map, so $\Phi_t(x)$ is the location of a particle at time $t$ if it is at $x$ at time 0.
- $v_t : \mathbf{R}^d \to \mathbf{R}^d$, the velocity field of the flow at time $t$.
- $\mu_t \in \mathcal{P}(\mathbf{R}^d)$, the probability measure at time $t$ with sample access and density $p_t$.

We will assume all these quantities are sufficiently continuous and bounded to ensure the Eulerian and Lagrangian viewpoints in Section 3.1 are equivalent. This can be achieved by using continuously differentiable activation functions in the network architectures and assuming the network weights are finite similar to the uniqueness arguments given in [Chen et al., 2018]. We can now parameterize the flow modeling either the flow map $\Phi_t$ or the velocity field $v_t$ as a neural network.

**Time-dependent Invertible Push Forward (TIPF).** We first parameterize a probability flow by modeling $\Phi_t : \mathbf{R}^d \to \mathbf{R}^d$ as an invertible network for every $t$. The network architecture is chosen so that $\Phi_t$ has an analytical inverse with a tractable Jacobian determinant, similar to [Biloš et al., 2021]. We augment RealNVP from Dinh et al. [2016] so that the network for predicting scale and translation takes $t$ as an additional input. To enforce the initial condition, we need $\Phi_0$ to be the identity map. This condition can be baked into the network architecture [Biloš et al., 2021] or enforced by adding an additional loss term $\mathbf{E}_{X \sim \mu_0^*} \|\Phi_0(X) - X\|^2$. For brevity, we will from now on omit in the text this additional loss term. The velocity field can be recovered via $v_t(x) = \partial_t \Phi_t(\Phi_t^{-1}(x))$. To recover the density $p_t$ of $\mu_t = (\Phi_t)_\# \mu_0$, we use the change-of-variable formula $\log p_t(x) = \log p_0^*(\Phi_t^{-1}(x)) + \log \det |J \Phi_t^{-1}(x)|$ (see (1) in Dinh et al. [2016]).

**Neural ODE (NODE).** We also parameterize a flow by modeling $v_t : \mathbf{R}^d \to \mathbf{R}^d$ as a neural network; this is used in Neural ODE [Chen et al., 2018]. The network only needs to satisfy the minimum requirement of being continuous. The flow map and the density can be recovered via numerical integration: $\Phi_t(x) = x + \int_0^t v_s(\Phi_s(x)) \, \mathrm{d}s$ and $\log p_t(\Phi_t(x)) = \log p_0^*(x) - \int_0^t \boldsymbol{\nabla} \cdot v_s(\Phi_s(x)) \, \mathrm{d}s$, a direct consequence of (1) also known as the instantaneous change-of-variable formula [Chen et al., 2018]. To obtain the inverse of the flow map, we integrate along $-v_t$. With NODE, the initial condition $\mu_0 = \mu_0^*$ is obtained for free.

We summarize the advantages and disadvantages of TIPF and NODE as follows. While the use of invertible coupling layers in TIPF allows exact access to samples and densities, TIPF becomes less effective in higher dimensions as many couple layers are needed to retain good expressive power, due to the invertibility requirement. In contrast, NODE puts little constraints on the network architecture, but numerical integration can have errors. Handling the initial condition is trivial for NODE while an additional loss term or special architecture is needed for TIPF. As we will show in the experiments, both strategies have merits.

## 3.3 Formulation

We now describe our algorithm for solving mass-conserving PDEs (2). A PDE of this form is determined by $f_t(\cdot; \mu_t) : \mathbf{R}^d \to \mathbf{R}^d$ plus the initial condition $\mu_0^*$. If a probability flow $\mu_t$ with flow map $\Phi_t$ and velocity field $v_t$ satisfies the following *self-consistency* condition,

$$v_t(x) = f_t(x; \mu_t), \forall x \text{ in the support of } \mu_t, \tag{6}$$

then the continuity equation of this flow implies the corresponding PDE (2) is solved. Conversely, the velocity field of any solution of (2) will satisfy (6). Hence, instead of solving (2) which is a condition on the density $p_t$ that might be hard to access, we can solve (6) which is a more tractable condition. Shen et al. [2022], Shen and Wang [2023] develop this concept for the Fokker-Planck equation and McKean-Vlasov type PDEs; here we generalize it to a wider class of PDEs of the form (2).

Let $\theta$ be the network weights that parameterize the probability flow using TIPF or NODE. The flow's measure, velocity field, and flow map at time $t$ are denoted as $\mu_t^\theta, v_t^\theta, \Phi_t^\theta$ respectively. One option to solve (6) would be to minimize the *self-consistency loss*

$$\min_\theta \int_0^T \mathbf{E}_{X \sim \mu_t^\theta} \left[ \left\| v_t^\theta(X) - f_t(X; \mu_t^\theta) \right\|^2 \right] \mathrm{d}t. \tag{7}$$

This formulation is reminiscent of PINNs [Raissi et al., 2019] where a residual of the original PDE is minimized. Direct optimization of (7) is challenging: while the integration over $[0, T]$ and $\mu_t^\theta$ can be approximated using Monte Carlo, to apply stochastic gradient descent, we must differentiate through $\mu_t^\theta$ and $f_t$: this can be either expensive or intractable depending on the network parameterization. The algorithm by Shen and Wang [2023] minimizes (7) with the adjoint method specialized to Fokker-Planck equations and McKean-Vlasov type PDEs; extending their approach to more general PDEs requires a closed-form formula for the time evolution of the quantities within $f_t$, which at best can only be obtained on a case-by-case basis.

Instead, we propose the following iterative optimization algorithm to solve (7). Let $\theta_k$ denote the network weights at iteration $k$. We define iterates

$$\theta_{k+1} := \theta_k - \eta \nabla_\theta|_{\theta=\theta_k} F(\theta, \theta_k), \tag{8}$$

where

$$F(\theta, \theta_k) := \int_0^T \mathbf{E}_{X \sim \mu_t^{\theta_k}} \left[ \left\| v_t^\theta(X) - f_t(X; \mu_t^{\theta_k}) \right\|^2 \right] dt. \tag{9}$$

Effectively, in each iteration, we minimize (9) by one gradient step where we match the velocity field $v_t^\theta$ to what it should be according to $f_t$ based on the network weights $\theta_k$ from the previous iteration. This scheme can be interpreted as a gradient descent on 7 using the biased gradient estimate $\nabla_\theta F(\theta, \theta_k)$—see Appendix A for a discussion. We call this iterative algorithm *self-consistent velocity matching*.

If $f_t$ depends on the density of $\mu_t$ only through the score $\nabla \log p_t$ (corresponding to a diffusion term in the PDE), then we can apply an integration-by-parts trick [Hyvärinen and Dayan, 2005] to get rid of this density dependency by adding a divergence term of the velocity field. Suppose $f_t$ is from the Fokker-Planck equation (4). Then the cross term in (9) after expanding the squared norm has the following alternative expression.

**Proposition 3.1.** *For every $t \in [0, T]$, for $f_t$ defined in (4), assume $v_t^\theta, D_t$ are bounded and continuously differentiable, and $\mu_t^{\theta'}$ is a measure with a continuously differentiable density $p_t^{\theta'}$ that vanishes in infinity and not at finite points. Then we have*

$$\mathbf{E}_{X \sim \mu_t^{\theta'}} \left[ v_t^\theta(X)^\top f_t(X; \mu_t^{\theta'}) \right] = \mathbf{E}_{X \sim \mu_t^{\theta'}} \left[ v_t^\theta(X)^\top b_t(X) + \boldsymbol{\nabla} \cdot \left( D_t^\top(X) v_t^\theta(X) \right) \right]. \tag{10}$$

We provide the derivation in Appendix B. With Proposition 3.1, we no longer need access to $\nabla \log p_t$ when computing $\nabla_\theta F$. This is useful for NODE parameterization since obtaining the score would otherwise require additional numerical integration.

Our algorithm is summarized in Algorithm 1 in the appendix. We use Adam optimizer [Kingma and Ba, 2014] to modulate the update (8). For sampling time steps $t_1, \ldots, t_L$ in $[0, T]$, we use stratified sampling where $t_l$ is uniformly sampled from $[(l-1)T/L, lT/L]$; such a sampling strategy results in more stable training in our experiments. We implemented our method using JAX [Bradbury et al., 2018] and FLAX [Heek et al., 2020]. See Appendix C for the implementation details.

## 4 Experiments

We show the efficiency and accuracy of our method on several PDEs of the form (2). We start with three Wasserstein gradient flow experiments (Section 4.1, Section 4.2, Section 4.3). Next, we consider the time-dependent Fokker-Planck equation that simulates attraction towards a harmonic mean in Section 4.4. Finally, in Section 4.5, we apply our framework to generate complicated low-dimensional dynamics including flows splashing against obstacles and smooth interpolation of measures. We will use SCVM-TIPF and SCVM-NODE to denote our method with TIPF and NODE parameterization respectively. We use JKO-ICNN to denote the method by Mokrov et al. [2021], JKO-ICNN-PD to denote the method by Fan et al. [2021] (PD for "primal-dual"), ADJ to denote the adjoint method by Shen and Wang [2023], SDE-EM to denote the Euler-Maruyama method for solving the SDE associated with the Fokker-Planck equation, and DFE ("discrete forward Euler") to denote the method by Boffi and Vanden-Eijnden [2023]. We implemented all competing methods in JAX—see more details in Appendix C—and we compare quantitatively against these methods when possible.

In Table 1, we compare the time complexity of training the described methods, where we show that SCVM-TIPF and SCVM-NODE have low computational complexity among all methods.

**Evaluation metrics.** For quantitative evaluation, we use the following metrics. To compare measures with density access, following Mokrov et al. [2021], we use the symmetric Kullback-Leibler (symmetric KL) divergence, defined as $\mathrm{SymKL}(\rho_1, \rho_2) := \mathrm{KL}(\rho_1 \parallel \rho_2) + \mathrm{KL}(\rho_2 \parallel \rho_1)$, where $\mathrm{KL}(\rho_1 \parallel \rho_2) := \mathbf{E}_{X \sim \rho_1}[\log \mathrm{d}\rho_1(X)/\mathrm{d}\rho_2(X)]$. Sample estimates of KL can be negative which complicates log-scale plotting, so when this happens, we consider an alternative $f$-divergence $D_f(\rho_1 \parallel \rho_2) := \mathbf{E}_{X \sim \rho_2}[(\log \rho_1(X) - \log \rho_2(X))^2/2]$ whose sample estimates are always non-negative. We similarly define the symmetric $f$-divergence $\mathrm{Sym}D_f(\rho_1, \rho_2) := D_f(\rho_1 \parallel \rho_2) + D_f(\rho_2 \parallel \rho_1)$. For particle-based methods, we use kernel density estimation (with Scott's rule) to obtain the density function before computing symmetric KL or $f$-divergence. We also consider the Wasserstein-2 distance [Bonneel et al., 2011] and the Bures-Wasserstein distance [Kroshnin et al., 2021]; these two measures only require sample access. All metrics are computed using i.i.d. samples. See Appendix C.6 for more details.

## 4.1 Sampling from mixtures of Gaussians

We consider computing the Wasserstein gradient flow of the KL divergence $\mathcal{F}(\mu) = \mathrm{KL}(\mu \parallel \mu^*)$ where we have density access to the target measure $\mu^*$. To fit into our framework, we set $f_t(x; \mu_t) = \nabla \log p^*(x) - \nabla \log p_t(x)$ which matches (4) with $b_t(x) = \nabla \log p^*(x)$ and $D_t(x) = I_d$. Following the experimental setup in Mokrov et al. [2021] and Fan et al. [2021], we take $\mu^*$ to be a mixture of 10 Gaussians with identity covariance and means sampled uniformed in $[-5, 5]^d$. The initial measure is $\mu_0^* = \mathcal{N}(0, 16I_d)$. We solve the corresponding Fokker-Planck PDE for a total time of $T = 5$ and for $d = 10, \ldots, 60$. As TIPF parameterization does not scale to high dimensions, we only consider SCVM-NODE in this experiment.

Figure 1 shows the probability flow produced by SCVM-NODE in dimension 60 at different time steps; as we can see, the flow quickly converges to the target distribution.

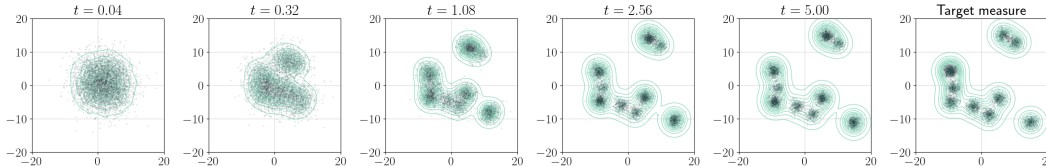

Figure 1: Probability flow produced by SCVM-NODE for a 60-dimensional mixture of Gaussians at irregular time steps. Samples are projected onto the first two PCA components and kernel density estimation is used to generate the contours.

In Figure 2, we quantitatively compare our method with Mokrov et al. [2021], Fan et al. [2021], and Shen and Wang [2023]. Training time is reported for all methods.

- In the left two columns of Figure 2, we find that even though the adjoint method ADJ [Shen and Wang, 2023] minimizes the self-consistency loss (7) directly, the decay of self-consistency can be much slower than that of SCVM-NODE as the dimension increases. We suspect this is due to the amount of error accumulated in the adjoint method which involves two numerical integration passes to obtain the gradient. Moreover, ADJ requires up to *third-order spatial derivatives* of the parameterized neural velocity field which can be inaccurate even if the consistency loss is low—in comparison SCVM-NODE only requires one integration pass and the first-order spatial derivative of the network. Despite the bias of the gradient used in SCVM-NODE, it finds more efficient gradient trajectories than ADJ. Additionally, ADJ takes 80 times longer to train than SCVM-NODE in dimension 10, and scaling up to higher dimensions becomes prohibitive.
- The rightmost column of Figure 2 shows SCVM-NODE achieves far lower symmetric KL compared to the JKO methods. The gradient flow computed by JKO methods does not decrease KL divergence monotonically, likely because the optimization at each JKO step has yet to reach the minimum even though we use 2000 gradient updates for each step. For both JKO methods, the running time for each JKO step increases linearly because samples (and for JKO-ICNN also $\log \det$ terms) need to be pushed through a growing chain of ICNNs; as a result, the total running time scales *quadratically* with the number of JKO steps. JKO methods also take about 40 times as long evaluation time as

SCVM-NODE in dimension 60 since density access requires solving an optimization problem for every JKO step. On top of the computational advantage and better results, our method also does not have temporal discretization: after being trained, the flow can be accessed at any time $t$ (Figure 1).

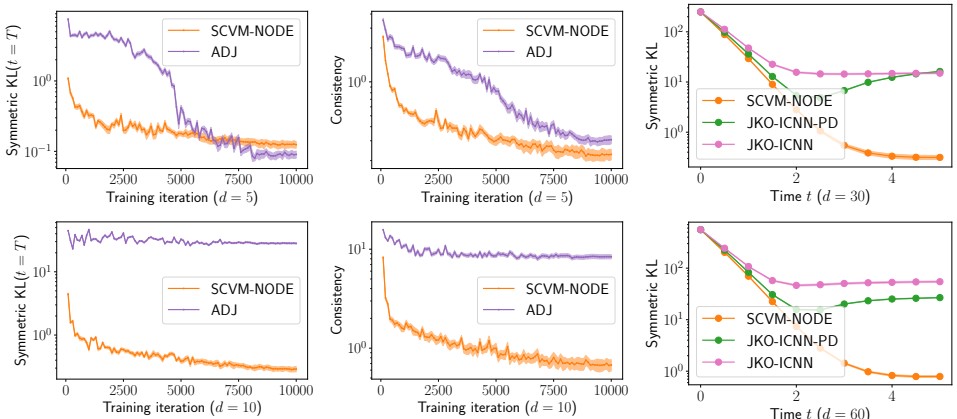

Figure 2: Quantitative comparison for the mixture of Gaussians experiment. The left two columns plot the symmetric KL (at $t = T$ compared against the target measure) and consistency (7) versus the training iterations for SCVM-NODE (ours) and ADJ [Shen and Wang, 2023]. The rightmost column plots the symmetric KL across time $t$ (compared against the target measure) for SCVM-NODE and the JKO methods in high dimensions. Training time: for $d = 10$, SCVM-NODE takes 7.37 minutes, ADJ takes 585.2 minutes; for $d = 60$, SCVM-NODE takes 23.9 minutes, JKO-ICNN takes 375.2 minutes, and JKO-ICNN-PD takes 24.4 minutes.

## 4.2 Ornstein-Uhlenbeck process

We consider the Wasserstein gradient flow of the KL divergence with respect to a Gaussian with the initial distribution being a Gaussian, following the same experimental setup in Mokrov et al. [2021], Fan et al. [2021]. In this case, the gradient flow at time $t$ is a Gaussian $G(t)$ with a known mean and covariance; see Appendix D.1 for details. We quantitatively compare all methods in Figure 3:

- ADJ achieves the best results in dimensions $d = 5$ and $d = 10$. However, this is at the cost of high training time: in dimension 10, ADJ takes 341 minutes to train, while SCVM-TIPF and SCVM-NODE take 23 and 9 minutes respectively for the same 20k training iterations. As such, we omit ADJ in higher-dimensional comparisons.
- Both our SCVM-TIPF and SCVM-NODE achieve good results second only to ADJ in dimensions 5 and 10. In low dimensions, SCVM-TIPF results in lower probability divergences than SCVM-NODE likely due to having exact density access. Although not shown, SCVM-TIPF also satisfies the initial condition well (numbers at $t = 0$ are comparable to those at $t = 0.25$ in the left two columns in Figure 3).
- For the two JKO methods, they result in much higher errors for $t \leq 0.5$ compared to later times: this is expected because the dependency of $G(t)$ on $t$ is exponential, so convergence to $\mu^*$ is faster in the beginning, yet a constant step size is used for JKO methods.
- For DFE, the result is highly sensitive to the forward Euler step size $\Delta t$. We choose step size $\Delta t = 0.01$ which empirically gives the best results among $\{0.1, 0.01, 0.001, 0.0001\}$. As DFE achieves far lower symmetric KL divergence or $f$-divergence compared to alternatives, we only include its Bures-Wasserstein distance in high dimensions where its number is slightly worse than alternatives (bottom right plot of Figure 3).

## 4.3 Porous medium equation

Following Fan et al. [2021], we consider the porous medium equation with only diffusion: $\partial_t p_t = \Delta p_t^m$ with $m > 1$. Its solution is the Wasserstein gradient flow of $\mathcal{F}(\mu) = \int \frac{1}{m-1} p(x)^m \, dx$ where $p$ is the density of $\mu$ with $\nabla_{W_2} \mathcal{F}(\mu)(x) = \nabla(\frac{m}{m-1} p^{m-1}(x))$—see Appendix D.2 for details. We consider only SCVM-TIPF and JKO methods here.

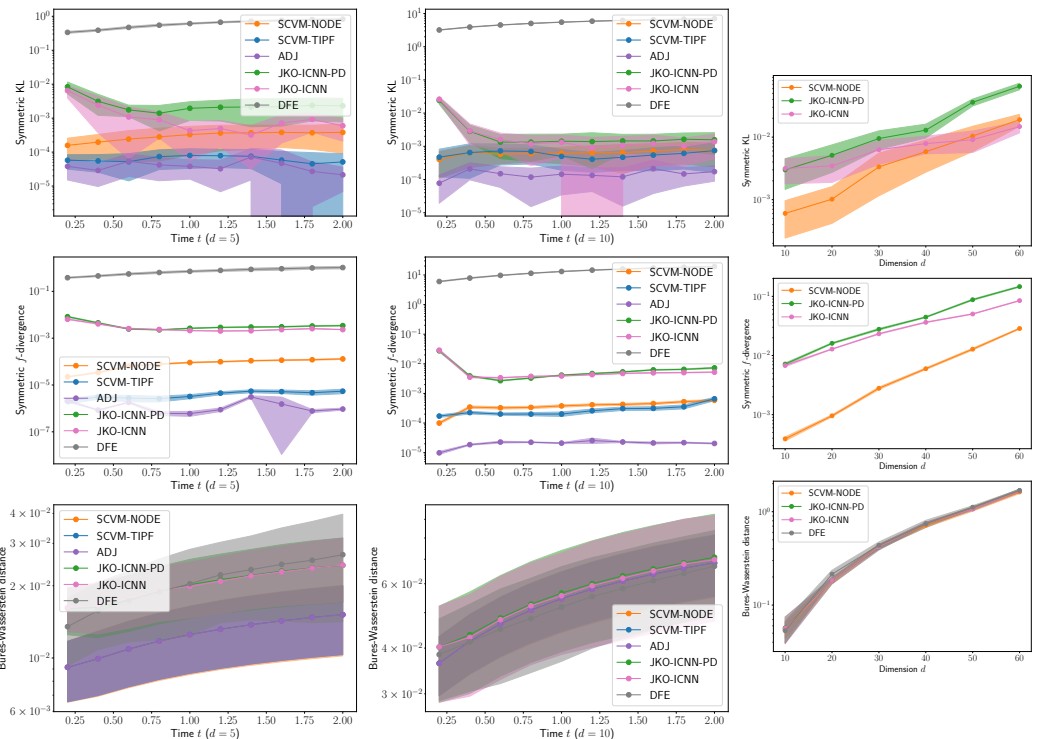

Figure 3: Quantitative results for the OU process experiment. The left two columns show the metrics (symmetric KL, symmetric $f$-divergence, and Bures-Wasserstein distance) versus time $t$ of various methods computed against the closed formed solution $G(t)$ in dimension $d = 5, 10$. The right column shows the metrics averaged across $t$ versus dimension $d$ in higher dimensions.

We show the efficiency of SCVM-TIPF compared to JKO-ICNN in dimension $d = 1, 2, \ldots, 6$. We exclude JKO-ICNN-PD because it produces significantly worse results. We visualize the density $p_t$ of the solution from SCVM-TIPF and JKO-ICNN on the top of Figure 4 in dimension 1 compared to $p_t^*$. Both methods approximate $p_t^*$ well with SCVM-TIPF being more precise at small $t$; this is consistent with the observation in Figure 2 where JKO methods result in bigger errors for small $t$.

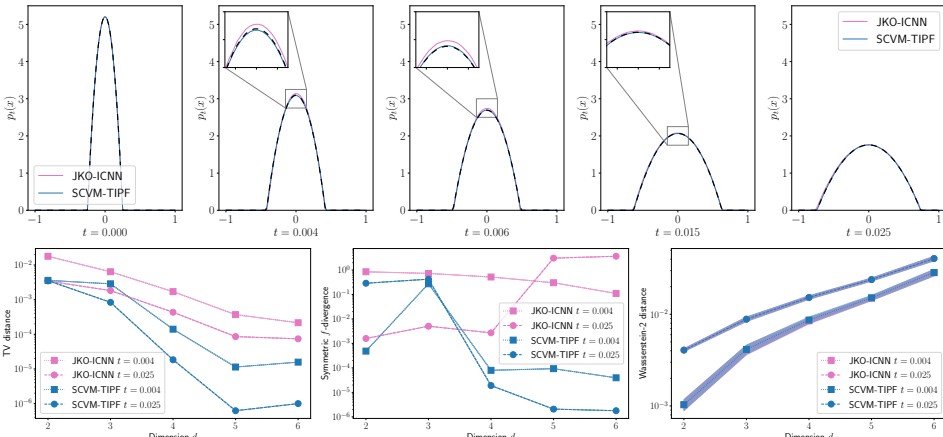

Figure 4: Top: visualization of the densities of $p_t^*$ and $p_t$ for the porous medium equation in dimension 1 at varying time steps $t$ for SCVM-TIPF and JKO-ICNN. Bottom: total variation distance, symmetric $f$-divergence, and Wasserstein-2 distances across dimensions at $t = 0.004$ and $t = 0.025$ between $p_t$ and $p_t^*$ for solving the porous medium equation.

On the bottom row of Figure 4, we plot the total variation (TV) distance, the symmetric $f$-divergence, and the Wasserstein-2 distance (details on the TV distance are given in Appendix C.6) between the recovered solution $p_t$ and $p_t^*$ for both methods at $t = 0.004$ and $t = 0.025$. Note that the values of all metrics are very low implying that the solution from either method is very accurate, with SCVM-TIPF more precise in TV distance and symmetric $f$-divergence, especially for $d > 3$. Like with the experiments in previous sections, JKO-ICNN is much slower to train: in dimension 6, training JKO-ICNN took 102 minutes compared to 21 minutes for SCVM-TIPF.

## 4.4 Time-Dependent Fokker-Planck equation

In this section, we qualitatively evaluate our method for solving a PDE that is not a Wasserstein gradient flow. In this case, JKO-based methods cannot be applied. Consider the OU process from Section 4.2 when the mean $\beta$ and the covariance matrix $\Gamma$ become time-dependent as $\beta_t$ and $\Gamma_t$. The resulting PDE is a time-dependent Fokker-Planck equation of the form (4) with

$$f_t(X, \mu_t) = \Gamma_t(\beta_t - X) - D\nabla \log p_t(X). \tag{11}$$

In this configuration, when the initial measure $p_0$ is Gaussian, the solution $\mu_t$ can again be shown to be Gaussian with mean and covariance following an ODE—see Appendix D.3 for more details. We consider, in dimension 2 and 3, time-dependent attraction towards a harmonic mean $\beta_t = a(\sin(\pi\omega t), \cos(\pi\omega t))$ using the expression of $\beta_t$ from Boffi and Vanden-Eijnden [2023], augmented to $\beta_t = a(\sin(\pi\omega t), \cos(\pi\omega t), t)$ in dimension 3.

We apply both SCVM-TIPF and SCVM-NODE to this problem and compare our results with alternatives. Similar to Figure 3, as shown in Figure 5, both SCVM-TIPF and SCVM-NODE achieve results on par with ADJ, with both SCVM methods being 30 times faster than ADJ in dimension 10. DFE results in good Wasserstein-2 metrics but worse divergences. Visualization of the evolution of a few sampled particles are given in Figure 9 and Figure 10.

In Appendix D.4, we augment (11) with an interaction term to simulate a flock of (infinitely many) birds, resulting in a non-Fokker-Planck PDE that can be readily solved by our method.

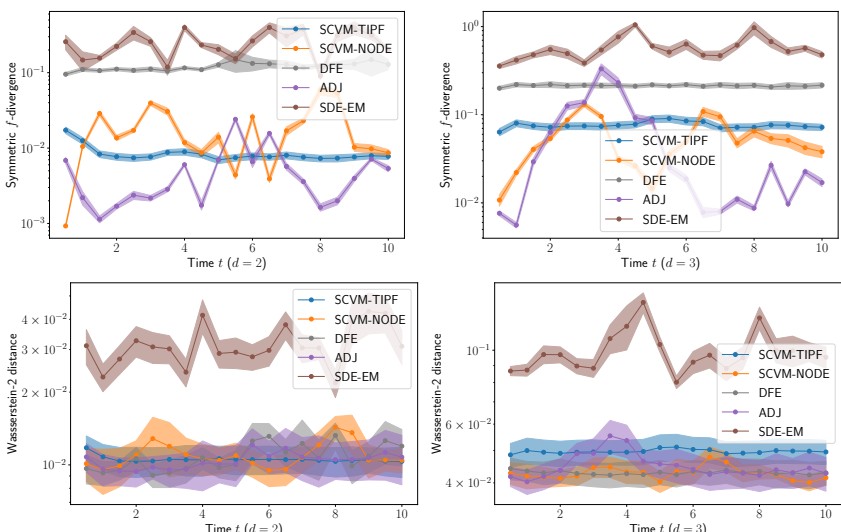

Figure 5: Symmetric KL divergence and Wasserstein-2 distances across time for $d = 2, 3$ between the recovered flows and the ground truth for the time-dependent Fokker-Planck equation.

## 4.5 Additional qualitative low-dimensional dynamics

To demonstrate the flexibility of our method, we apply our algorithm to model more general mass-conserving dynamics than the ones considered in the previous sections.

**Flow splashing against obstacles.** We model the phenomenon of a 2-dimensional flow splashing against obstacles using a Fokker-Planck equation (4) where $b_t$ encodes the configuration of three

obstacles that repel the flow (See Appendix D.5 for details). We solve this PDE using SCVM-NODE for $T = 5$ and visualize the recovered flow in (6). When solving the same PDE using SDE-EM, the flow incorrectly crosses the bottom right obstacle due to a finite time step size (Figure 14). When using DFE, the path of initial samples appears jagged (right of Figure 13); our method has no such issue and results in continuous sample paths (left of Figure 13). Method ADJ suffers from numerical instability and cannot be trained without infinite loss in this example.

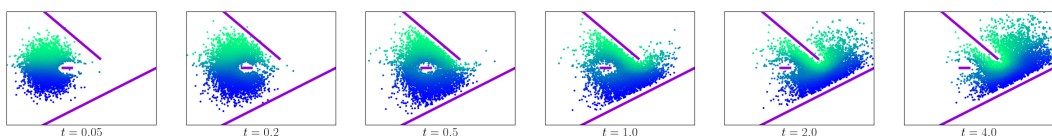

Figure 6: A flow splashing against three obstacles (in purple) produced by SCVM-NODE. Particles are colored based on the initial $y$ coordinates.

**Smooth interpolation of measures.** To illustrate the flexibility of our method, we demonstrate two ways to formulate the problem of smoothly interpolating a list of measures. First, we model the interpolation as a time-dependent Fokker-Planck equation and use it to interpolate MNIST digits 1, 2, and 3, starting from a Gaussian (Figure 7). Next, we adopt an optimal transport formulation and use it to generate an animation sequence deforming a 3D hand model to a different pose and then to a ball, similar to the setup in Zhang et al. [2022]. Note that the optimal transport formulation is not solvable using competing methods. See Appendix D.6 for more details.

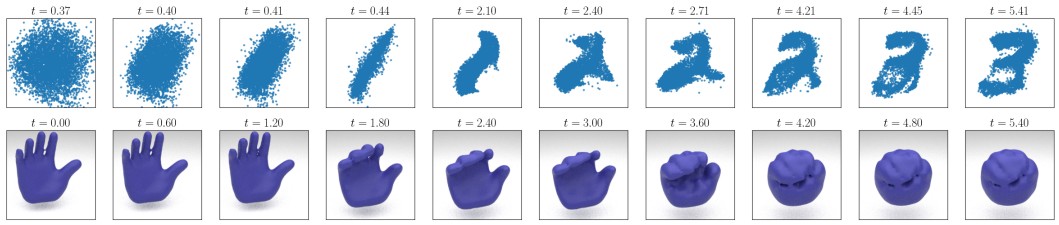

Figure 7: Smooth interpolation of measures. Top: interpolating MNIST digits 1 to 3. Bottom: interpolating hand from the initial pose to a different pose and then to a ball.

# 5   Conclusion

By extending the concept of self-consistency from Shen et al. [2022], we present an iterative optimization method for solving a wide class of mass-conserving PDEs without temporal or spatial discretization. In all experiments considered, our method achieves strong quantitative results with significantly less training time than JKO-based methods and the adjoint method in high dimensions.

Below we highlight a few future directions. First, as discussed, the two ways to parameterize a probability flow, TIPF, and NODE, both have their specific limitations. Finding a new parameterization that combines the advantages of both TIPF and NODE is an important next step. Secondly, we hope to extend our approach to incorporate more complicated boundary conditions. Finally, given that the proposed algorithm is highly effective empirically, it would be an interesting theoretical step to explore its convergence properties.

**Acknowledgements**   The MIT Geometric Data Processing group acknowledges the generous support of Army Research Office grants W911NF2010168 and W911NF2110293, of Air Force Office of Scientific Research award FA9550-19-1-031, of National Science Foundation grant CHS-1955697, from the CSAIL Systems that Learn program, from the MIT–IBM Watson AI Laboratory, from the Toyota–CSAIL Joint Research Center, from a gift from Adobe Systems, and from a Google Research Scholar award.

SH acknowledges the financial support from the University of Bordeaux (UBGR grant) and the French Research Agency (PostProdLEAP).

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

| SCVM-TIPF | SCVM-NODE | ADJ | JKO-ICNN | JKO-ICNN-PD |
|:---:|:---:|:---:|:---:|:---:|
| $O(STd^2)$ | $O(STN_{\text{ode}}d)$ | $O(STN_{\text{ode}}d^3)$ | $O(ST^2d^3)$ | $O(ST^2d)$ |

Table 1: Time complexity of training for $S$ iterations for the methods considered, in terms of dimension $d$. We assume for simplicity that the same batch size is used, the training is done for $T$ time steps, and any network forward pass takes $O(d)$ time. For SCVM-NODE and ADJ, $N_{\text{ode}}$ denotes the number of ODE integration steps. For SCVM-TIPF, RealNVP is used to build the coupling layer, and we use $d$ coupling layers, hence the extra multiple of $d$. For ADJ, the $d^3$ term comes from the third-order spatial derivatives. For JKO-ICNN, the $d^3$ term is due to computing the log-determinant term. For both JKO methods, the quadratic dependency on $T$ is due to maintaining a growing chain of neural networks of size $T$ as described in the related works section.

# Appendix

In this appendix, we provide details and further justification of the proposed method. In Appendix A, we provide an interpretation of the update (9) as a gradient descent step with a biased gradient. In Appendix B, we explain the integration-by-parts trick used to prove (3.1). In Appendix C, we provide implementation details of all considered methods. In Appendix D, we provide additional experimental details and results.

---

**Algorithm 1** Self-consistent velocity matching

---

   **Input:** $f_t(\cdot,\cdot), \mu_0^*, T, N_{\text{train}}, B, L$.
   Initialize network weights $\theta$.
   **for** $k = 1, \ldots, N_{\text{train}}$ **do**
      $\theta' \leftarrow \theta$.
      Sample $x_1, \ldots, x_B \sim \mu_0^*, t_1, \ldots, t_L \sim [0, T]$.
      $y_{b,l} \leftarrow \Phi_{t_l}^{\theta'}(x_b), \forall b = 1, \ldots, B, l = 1, \ldots, L$.
      $\theta \leftarrow \theta - \eta \nabla_\theta \frac{1}{BL} \sum_{b,l} \left\| v_{t_l}^\theta(y_{b,l}) - f_{t_l}(y_{b,l}; \mu_{t_l}^{\theta'}) \right\|^2$.
   **end for**
   **Output:** optimized $\theta$.

---

## A  Biased Gradient Interpretation

Assume $\mu_t^\theta$ has density $p_t^\theta$. Using $f_t^\theta(x)$ to denote $f_t(x; \mu_t^\theta)$ from (7), the self-consistency loss can be written as,

$$L(\theta) := \int_0^T \int p_t^\theta(x) \left\| v_t^\theta(x) - f_t^\theta(x) \right\|^2 \mathrm{d}x \, \mathrm{d}t.$$

Assuming all terms involving $\theta$ are differentiable with respect to $\theta$, the gradient of $L(\theta)$ with respect to the neural network parameters $\theta$ can be written as:

$$\nabla L(\theta) = \int_0^T \int \nabla_\theta p_t^\theta(x) \left\| v_t^\theta(x) - f_t^\theta(x) \right\|^2 \mathrm{d}x \, \mathrm{d}t \tag{12}$$

$$+ 2 \int_0^T \int p_t^\theta(x) J_\theta v_t^\theta(x)^\top (v_t^\theta(x) - f_t^\theta(x)) \, \mathrm{d}x \, \mathrm{d}t \tag{13}$$

$$- 2 \int_0^T \int p_t^\theta(x) J_\theta f_t^\theta(x)^\top (v_t^\theta(x) - f_t^\theta(x)) \, \mathrm{d}x \, \mathrm{d}t. \tag{14}$$

Here we use $J_\theta$ to denote the Jacobian with respect to $\theta$. On the other hand, the gradient used in the updates (8) is $\nabla_\theta F(\theta, \theta')$ at $\theta' = \theta$:

$$\nabla_\theta F(\theta, \theta') \Big|_{\theta' = \theta} = 2 \int_0^T \int p_t^\theta(x) J_\theta v_t^\theta(x)^\top (v_t^\theta(x) - f_t^\theta(x)) \, \mathrm{d}x \, \mathrm{d}t. \tag{15}$$

We see that (15) is exactly the middle term (13). Hence our formulation can be interpreted as doing gradient descent with a biased gradient estimator. It remains a future work direction to

theoretically analyze the amount of bias in (15) and the condition under which the dot product $\langle \nabla L(\theta), \nabla_\theta F(\theta, \theta')|_{\theta'=\theta}\rangle \geq 0$. The central challenge would be to relate $J_\theta v_t^\theta$ and $J_\theta f_t^\theta$; this depends on the neural network architecture and the type of the PDE.

## B    Integration-by-Parts Trick

This is a common trick used in score-matching literature [Hyvärinen and Dayan, 2005].

*Proof of Proposition 3.1.* Fix $t > 0$. The form of $f_t$ in (4) is

$$f_t(x; \mu_t) = b_t(x) - D_t(x)\nabla \log p_t(x).$$

Hence

$$\mathbf{E}_{X \sim \mu_t^{\theta'}}\left[v_t^\theta(X)^\top f_t(X; \mu_t^{\theta'})\right] = \mathbf{E}_{X \sim \mu_t^{\theta'}}\left[v_t^\theta(X)^\top b_t(X)\right] - \mathbf{E}_{X \sim \mu_t^{\theta'}}\left[v_t^\theta(X)^\top D_t(X)\nabla \log p_t^{\theta'}(X)\right].$$

The second term can be written as

$$
\begin{aligned}
\mathbf{E}_{X \sim \mu_t^{\theta'}}\left[v_t^\theta(X)^\top D_t(X)\nabla \log p_t^{\theta'}(X)\right] &= \int v_t^\theta(x)^\top D_t(x)\nabla \log p_t^{\theta'}(x)\,\mathrm{d}p_t^{\theta'}(x) \\
&= \int v_t^\theta(x)^\top D_t(x)\nabla p_t^{\theta'}(x)/p_t^{\theta'}(x) \cdot p_t^{\theta'}(x)\,\mathrm{d}x \\
&= \int v_t^\theta(x)^\top D_t(x)\nabla p_t^{\theta'}(x)\,\mathrm{d}x \\
&= -\int \boldsymbol{\nabla}\cdot\left(D_t(x)^\top v_t^\theta(x)\right)p_t^{\theta'}(x)\,\mathrm{d}x \\
&= -\mathbf{E}_{X \sim \mu_t^{\theta'}}\left[\boldsymbol{\nabla}\cdot\left(D_t(X)^\top v_t^\theta(X)\right)\right],
\end{aligned}
$$

where we use integration-by-parts to get the second last equation and the assumption that $v_t^\theta, D_t$ are bounded and $p_t^\theta(x) \to 0$ as $\|x\| \to \infty$. ∎

## C    Implementation Details

### C.1    Network architectures for SCVM.

For TIPF, our implementation follows Dinh et al. [2016]. Each coupled layer uses 3-layer fully connected networks with layer sizes 64, 128, 128 for both scale and translation prediction. We use twice as many coupling layers as the dimension of the problem while each coupling layer updates one coordinate; we found using fewer layers with random masking gives much worse results.

For NODE, we use a 3-layer fully connected network for modeling the velocity field with layer size 256. We additionally add a low-rank linear skip connection $x \mapsto A(t)x + b(t)$ where $A(t) = L(t)L^\top(t)$ and $L(t)$ is a $d \times 20$ matrix to make $A(t)$ low-rank.

We use SILU activation [Elfwing et al., 2018] which is smooth for all layers for both TIPF and NODE. For NODE, we apply layer normalization before applying activation. We also add a sinusoidal embedding for the time input $t$ plus two fully connected layers of size 64 before concatenating it with the spatial input.

The numerical integration for NODE is done using Diffrax library [Kidger, 2021] with a relative and absolute tolerance of $10^{-4}$; we did not find considerable improvement when using a lower tolerance.

We use the integration-by-parts trick for SCVM-NODE whenever possible. Since TIPF has tractable log density, we do not use such a trick and optimize (9) directly for SCVM-TIPF which we found to produce better results.

### C.2    Hyperparameters.

Unless mentioned otherwise, we choose the following hyperparameters for Algorithm 1. We set $N_{\text{train}} = 10^5$ or $2 \times 10^5$, $B = 1000$, $L = 10$ or $20$. We use Adam [Kingma and Ba, 2014] with

a cosine decay learning rate scheduler, with initial learning rate $10^{-3}$, the number of decay steps same as $N_{\text{train}}$, and $\alpha = 0.01$ (so the final learning rate is $10^{-5}$). Since we are effectively performing gradient descent using a biased gradient, we set $b_2 = 0.9$ in Adam (instead of the default $b_2 = 0.999$), so that the statistics in Adam can be updated more quickly; we found this tweak improves the results noticeably.

### C.3 Implementation of JKO methods.

We base our JAX implementation of ICNN on the codebase by the original ICNN author: `https://github.com/facebookresearch/w2ot`. Compared to the original ICNN implementation by Amos et al. [2017], we add an additional convex quadratic skip connections used by Mokrov et al. [2021], which we found to be crucial for the OU process experiment. For ICNNs, we use hidden layer sizes 64, 128, 128, 64. The quadratic rank for the convex quadratic skip connections is set to 20. The activation layer is taken to be CELU.

To implement the method by Fan et al. [2021], we model the dual potential as a 4-layer fully connected network with layer size 128, with CELU activation. For the gradient flow of KL divergence and generalized entropy (used in Section 4.3), we follow closely the variational formulation and the necessary change of variables detailed in Fan et al. [2021, Corollary 3.3, Corollary 3.4].

In order to compute the log density at any JKO step, following Mokrov et al. [2021], we need to solve a convex optimization to find the inverse of the gradient of an ICNN. We use the LBFGS algorithm from JAXopt [Blondel et al., 2021] to solve the optimization with tolerance $10^{-2}$ (except for Section 4.3 we use a tolerance of $10^{-3}$ to obtain finer inverses, but it takes 6x longer compared to $10^{-2}$).

We always use 40 JKO steps, consistent with past works. For each JKO step, we perform 1000 stochastic gradient descent using Adam optimizer with a learning rate of $10^{-3}$, except for the mixture of Gaussians experiment, we use 2000 steps—using fewer steps will result in worse results. We have tested with the learning rate schedules used in Fan et al. [2021], Mokrov et al. [2021] and did not notice any improvement.

### C.4 Implementation of ADJ method.

We implement the adjoint method carefully following the formulation in Shen and Wang [2023]. The neural network for parameterizing the velocity field is identical to the one used in SCVM-NODE. The ODE integration also uses the same hyperparameters as that of SCVM-NODE. This way we can compare ADJ with SCVM-NODE in a fair manner since they only differ in the gradient estimation.

### C.5 Implementation of DFE method.

We implement DFE following the algorithm outlined in Boffi and Vanden-Eijnden [2023]. We use 5000 particles. For score estimation, we use the same network architecture as in NODE. At each time step, we optimize the score network 100 steps. We found the result of DFE depends tremendously on the time step size $\Delta t$. For the OU process experiment in dimension 60, when $\Delta t = 0.1, 0.01, 0.001, 0.0001$, the resulting Bures-Wasserstein distance at the final time to the target measure is $28.11, 0.31, 0.46, 9.21$ respectively. Surprisingly, a smaller $\Delta t$ can result in bigger errors. We choose $\Delta t = 0.01$ since it gives the best results.

### C.6 Evaluation metrics

For all our experiments, calculations of all metrics are repeated 20 times on 1000 samples from each distribution. Our plots show both the average and the standard deviation calculated over these 20 repetitions.

When estimating symmetric KL divergence using samples, due to the finite sample size and the numerical error in estimating the log density, the estimated divergence can be very close to zero or even negative (when this occurs we take absolute values). This explains why the standard deviation regions touch the $x$-axis in the log-scale plots in Figure 3.

To compute Bures-Wasserstein distance [Kroshnin et al., 2021], we first fit a Gaussian to the samples of either distribution and then compute the closed-form Wasserstein-2 distance between the two Gaussians.

For the porous medium equation (Section 4.3), the total variation distance is used in Figure 4 and Figure 8 to compare the estimated and ground-truth solutions. It is approximated by the $L_1$ distance between the densities calculated over $50000$ samples uniformly distributed on the compact $[-1.25x_{\max}, 1.25x_{\max}]$ with $x_{\max} = C/\left(\beta(t+t_0)^{\frac{-2\alpha}{d}}\right)$ being the bound of the support of $p_t^*$.

## D   Additional Experimental Details

### D.1   Ornstein-Uhlenbeck process

The OU process is the Wasserstein gradient flow of the KL divergence with respect to a Gaussian $\mu^* = \mathcal{N}(\beta, \Gamma^{-1})$ where $\beta \in \mathbf{R}^d$ and $\Gamma$ is a $d \times d$ positive-definite matrix. When the initial distribution is $\mu_0^* = \mathcal{N}(0, I_d)$, the gradient flow at time $t$ is known to be a Gaussian distribution $G(t)$ with mean $(I_d - e^{-t\Gamma})\beta$ and covariance $\Gamma^{-1}(I_d - e^{-2t\Gamma}) + e^{-2t\Gamma}$. We set the total time $T = 2$.

### D.2   Porous medium equation

This flow has as closed-form solution given by the Barenblatt profile Vázquez [2007] when initialized accordingly:

$$p_t^*(x) = (t + t_0)^{-\alpha} \left(C - \beta\|x\|^2(t+t_0)^{\frac{-2\alpha}{d}}\right)_+^{\frac{1}{m-1}}, \tag{16}$$

where $t_0 > 0$ is the starting time, $\alpha = \frac{m}{d(m-1)+2}$, $\beta = \frac{(m-1)\alpha}{2dm}$, and $C > 0$ is a free constant. Similar to Fan et al. [2021], we choose $m = 2$ and total time $T = 0.025$. The initial measure follows a Barenblatt distribution supported in $[-0.25, 0.25]^d$ ($C$ is chosen accordingly) with $t_0 = 10^{-3}$. We use Metropolis-Hastings to sample from $\mu_0$.

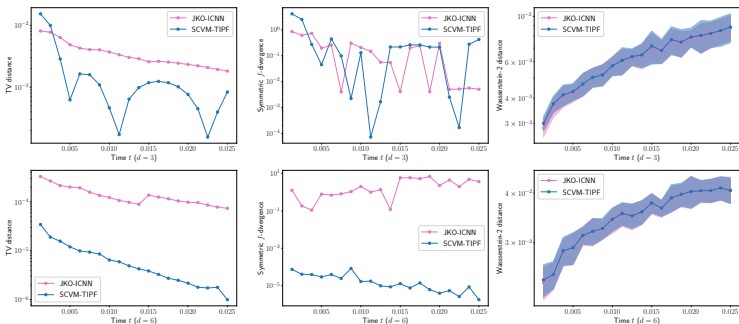

Figure 8: Metrics (TV, Symmetric $f$-divergence and Wasserstein-2 distance) across time for dimensions 3 and 6 between the estimated $\mu_t$ and the ground-truth $\mu_t^*$ when solving the Porous Medium Equation.

### D.3   Time-Dependant Fokker-Planck equation

We consider a time-dependent Fokker-Planck equation of the form (4) with the velocity field

$$f_t(X, \mu_t) = \Gamma_t(X - \beta_t) - D_t\nabla\log p_t(X). \tag{17}$$

When the initial measure $p_0$ is Gaussian, the solution $\mu_t$ can again be shown to be Gaussian with mean $m_t$ and covariance $\Sigma_t$ solutions of the differential equations:

$$\begin{cases} m_t' &= -\Gamma_t(m_t - \beta_t) \\ \Sigma_t' &= -\Gamma_t\Sigma_t - \Sigma_t\Gamma_t^\top + 2D_t. \end{cases} \tag{18}$$

In practice, we experiment with constant $\Gamma_t = \text{diag}(1, 3)$ and $D_t = \sigma^2 I_d$. We also experience in dimension 3 by considering and $\Gamma_t = \text{diag}(1, 3, 1)$. We set $a = 3$, $\omega = 1$, $\sigma = \sqrt{0.25}$ and pick as initial distribution $p_0$ a Gaussian with mean $b_0$ and covariance $\sigma^2 I_d$. We set the total time to $T = 10$.

We plot in Figure 9, for dimension 2, snapshots at different time steps of particles following the flow given by our method with TIPF parametrization. We only show SCVM-TIPF because SCVM-NODE gives visually indistinguishable trajectories. We also plot in Figure 10 the evolution of particles simulated by Euler-Maruyama (EM-SDE) discretization of the Fokker-Planck equation. Corresponding animated GIFs be found at this link.

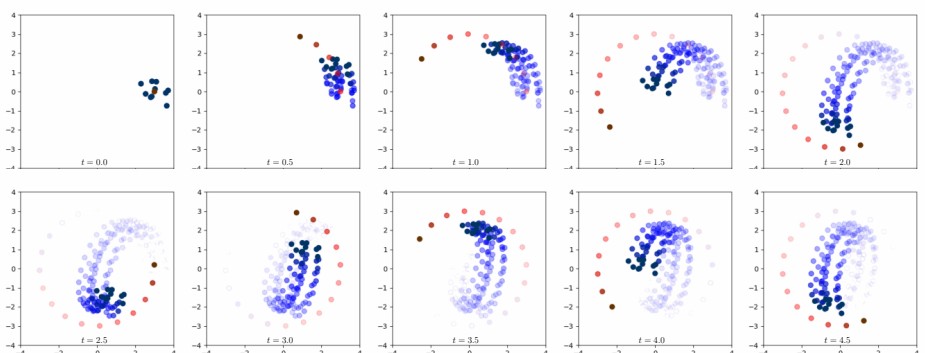

Figure 9: Evolution of particles (in blue) following the flow learned with SCVM-TIPF for the time-dependent OU process (Section 4.4). In red is the moving attraction trap.

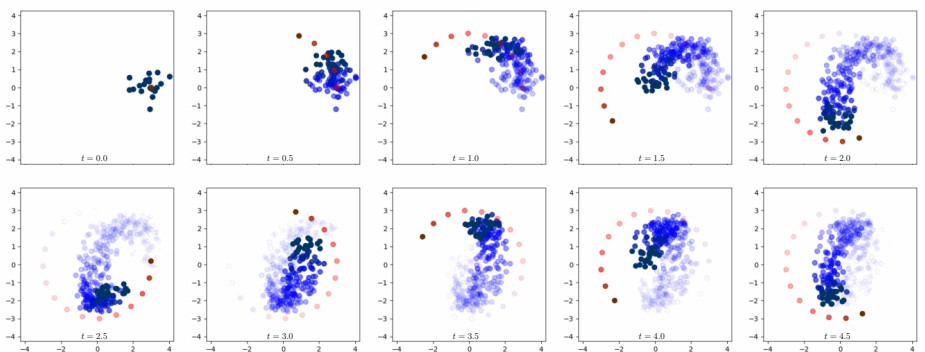

Figure 10: Evolution of particles (in blue) obtained by SDE-EM discretization for the time-dependent OU process (Section 4.4). In red is the moving attraction trap.

### D.4   Flock of birds

We model the dynamics of a flock of birds by augmenting the time-dependent Fokker-Planck equation (11) with an interaction term:

$$f_t(X, \mu_t) = \Gamma_t(\beta_t - X) + \alpha_t(X - \mathbf{E}[\mu_t]) - D\nabla \log p_t(X).$$

This is similar to the harmonically interacting particles experiment in Boffi and Vanden-Eijnden [2023], but we use a population expectation $\mathbf{E}[\mu_t]$ instead of an empirical one in modeling the repulsion from the mean. Since $f_t$ needs to access $\mathbf{E}[\mu_t]$, the resulting PDE is not a Fokker-Planck equation (4) and hence not solvable using the method in Boffi and Vanden-Eijnden [2023] but can be solved with our method by estimating $\mathbf{E}[\mu_t]$ using Monte Carlo samples from $\mu_t$. We use a similar setup as in Section 4.4, except we now use an "infinity sign" attraction $\beta_t = a(\cos(2\pi\omega t), 0.5\sin(2\pi\omega t))$ along with a sinusoidal $\alpha_t = 2\sin(\pi w t)$. Depending on the sign of $\alpha_t$, particles are periodically attracted towards or repulsed from their mean. Both SCVM-TIPF and SCVM-NODE produce similar visual results as shown in Figure 11 and Figure 12.

We use a constant $\Gamma_t = I_d$ and a constant diffusion matrix $D = \sigma^2 I_d$. We set $a = 3$, $\omega = 0.5$, and $\sigma = \sqrt{0.25}$. We pick as initial distribution $p_0$ a Gaussian with mean $(0, 0)$ and covariance $\sigma^2 I_d$. We set the total time to $T = 10$.

We respectively show in Figure 11 and Figure 12 simulations of particles following the flow learned with SCVM-TIPF and SCVM-NODE. Corresponding animated GIFs be found at this link.

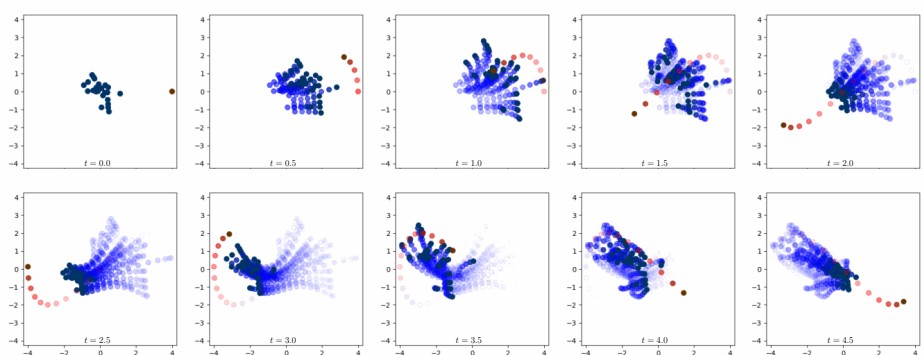

Figure 11: Evolution of particles following the flow trained with TIPF parametrization on the flock of birds PDE (Section 4.5). In red shows the moving attraction mean.

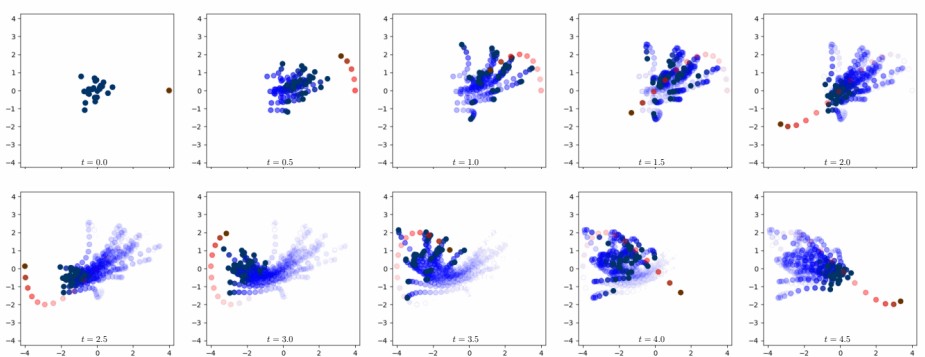

Figure 12: Evolution of particles following the flow trained with NODE parametrization on the flock of birds PDE (Section 4.5). In red shows the moving attraction mean.

### D.5 Flow splashing against obstacles

We use the following formulation for modeling the flow. Each obstacle is modeled as a line segment. The endpoints of the three obstacles are:

$$((0, 3), (3, 0.5)), ((1, 0), (1.5, 0)), ((-2, -4), (6, 0)).$$

We model the dynamics as a Fokker-Planck equation where $f_t$ of the form (4) is defined as

$$b_t(x) = (q_{\text{sink}} - x) + 20 \sum_{i=1}^{3} \frac{x - \pi_{O_i}(x)}{\|x - \pi_{O_i}(x)\|} p_{\mathcal{N}(0,0.04)}(\|x - \pi_{O_i}(x)\|),$$

$$D_t(x) = I_2,$$

where $q_{\text{sink}} = (4, 0)$, and $\pi_{O_i}(x)$ is the projection of $x$ onto obstacle $i$ represented as a line segment, and $p_{\mathcal{N}(0,0.04)}$ is the density of an 1-dimensional Gaussian with variance $0.04$.

The initial distribution is chosen to be $\mathcal{N}(0, 0.25I_2)$. We train SCVM-NODE for $10^4$ with an initial learning rate of $10^{-4}$. Training takes 5.4 minutes. The time step size for SDE-EM used to produce Figure 14 is 0.005. Corresponding animated GIFs be found at this link.

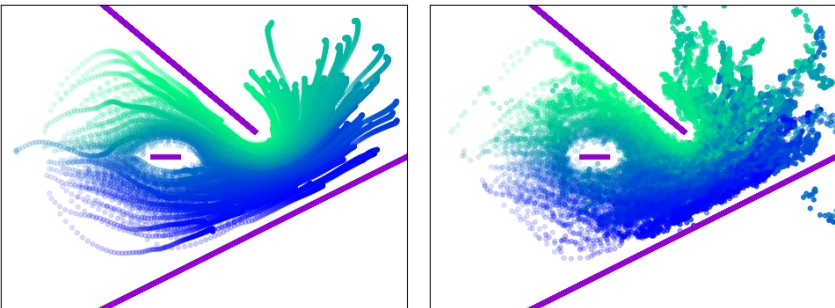

Figure 13: Trajectory of 200 random particles across time using the same setup as in Figure 6. Left are sample paths obtained by our method, and right are sample paths obtained by DFE [Boffi and Vanden-Eijnden, 2023].

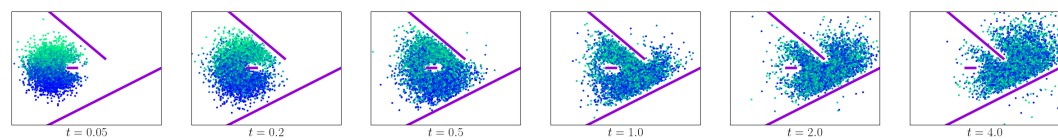

Figure 14: Same setup as in Figure 6 but with SDE-EM. We see the paths of the particles are not continuous. Moreover, the particles spill over the obstacle on the bottom right due to a finite time step size. In comparison, SCVM-NODE does not have such a problem.

### D.6 Smooth interpolation of measures

Suppose we are to smoothly interpolate $M$ measures $\nu_1, \ldots, \nu_M$ with densities $q_1, \ldots, q_M$, and we want the flow to approximate $\nu_i$ at time $r_i$. To achieve this goal, we present two formulations using different choices of $f_t$. We use SCVM-NODE in this section.

**Measure interpolation using time-dependent Fokker-Planck equations.** We model the dynamics as a Fokker-Planck equation where $f_t$ of the form (4) is taken to be

$$b_t(x) = \sum_{i=1}^{M} \phi(t - r_i)(\nabla \log q_i(x) - \nabla \log p_t(x))$$
$$D_t(x) = I_2,$$

where $\phi(t)$ is defined as the continuous bump function

$$\phi(t) = \begin{cases} 1.0 & |t| < 0.5h \\ (0.6h - |t|)/(0.1h) & |t| < 0.6h \\ 0.0 & \text{otherwise,} \end{cases}$$

for bandwidth $h = 1.0$.

Below we provide details for the MNIST interpolation result in the top row of Figure 7. We use the first three images of $1, 2, 3$ from the MNIST dataset. To construct $\nu_i$ from a digit image, we use a mixture of Gaussians where we put one equally-weighted Gaussian with covariance $0.02^2 I_2$ on the pixels with values greater than 0.5 (images are first normalized to have values in $[0, 1]$). The initial distribution is $\mathcal{N}((0.5, 0.5), 0.04 I_2)$. To train SCVM-NODE, we use an initial learning rate of $10^{-4}$ with cosine decay for a total of $10^5$ iterations. This takes roughly 1 hour to train.

**Measure interpolation using optimal transport.** An alternative way to interpolate measures using our framework is to use optimal transport to define $f_t$. Recall $\mu_t$ denotes the probability flow at time $t$. We then define

$$f_t(x) = \sum_{i=1}^{M} \phi(t - r_i)\nabla_{W_2} W_2^2(\mu_t, \nu_i),$$

where $W_2^2$ is the squared Wasserstein-2 distance and $\nabla_{W_2} W_2^2$ is its Wasserstein gradient. In practice, we compute $\nabla_{W_2} W_2^2$ using sample access and we employ the debiased Sinkhorn divergence [Genevay et al., 2018, Feydy et al., 2019] implemented in the JAX OTT library [Cuturi et al., 2022]. This formulation differs from the one in Zhang et al. [2022] in that here we prescribe the precise PDE based on $f_t$, whereas in Zhang et al. [2022] an optimal transport loss is used to fit the keyframes along with many regularizers on the velocity field $v_t$ to promote the smoothness and other desirable properties. In contrast, we do not use any regularizer on $v_t$.

To train SCVM-NODE to produce the hand-hand-ball animation sequence in the bottom row of Figure 7, we first sample 20000 points from the interior of the three meshes (a hand mesh, a hand mesh in a different pose, and a ball mesh), and we set $\nu_i$ to be the empirical measure of the corresponding point cloud. Note that different from the first formulation using Fokker-Planck equations, in the optimal transport formulation, throughout we only require sample access from each $\nu_i$. We use an initial learning rate of $10^{-4}$ with cosine decay for a total of $5 \times 10^4$ iterations. This takes 4.5 hours, which is significantly longer than the training time in Zhang et al. [2022] (reported to be 15 minutes). We leave further improvement of our method to interpolate measures faster as future work.

To render the animation, we sample 20000 points and render the point cloud at each time step using metaballs along with smoothing, similar to the procedure in Zhang et al. [2022]. We did not use the barycentric interpolation postprocessing step in Zhang et al. [2022] which makes sure the key measures $v_i$'s are fit exactly in the resulting animation. We also did not use unbalanced optimal transport, which as reported in Zhang et al. [2022] can make the fingers of the hand more separated, but requires careful parameter tuning.

