# OpenReview forum: "Self-Consistent Velocity Matching of Probability Flows"
_NeurIPS.cc/2023/Conference — NeurIPS 2023 poster_

### Official Review · Reviewer_egp4 · 2023-07-04

**Soundness:** 3 good
**Presentation:** 3 good
**Contribution:** 3 good
**Rating:** 5
**Confidence:** 3

**Summary:**

This paper proposes to solve a class of mass preserving PDEs by minimizing a loss defined with a self-consistency condition. Also, instead of minimizing this loss directly, an iterative formulation with a biased gradient estimator is proposed. Proposed method is a extension of two previous papers, extending their self-consistency to mass preserving PDEs, and getting rid of the usage of the expensive adjoint method.

**Strengths:**

The class of PDEs being solved is important, including the celebrated Fokker-Planck and Wasserstein gradient flow.

The computational complexity of this method is much lower than the previous ones, while preserving good empirical results.

Proposed method does not require spatial and temporal discretization, making it possible to use in high-dimensional case.

**Weaknesses:**

Although the mass preserving PDE (2) will always induce the self consistency condition (6), the other way around does not work, and it is not argued in much details when this will fail.

The proposed method does not require discretization, and it is motivated that the proposed method works better than the previous methods on high-dimensional cases, but very few experiments were performed in the high-dimensional environment.

**Questions:**

What is the computational complexity of the method, with regard to the dimension of the PDEs?

**Limitations:**

Proposed method only works for mass-preserving PDEs.

---

> ### Author Rebuttal · Authors · 2023-08-08
>
> Thank you for your valuable review. Please find below our response to your specific questions.
>
> **Although the mass-preserving PDE (2) will always induce the self-consistency condition (6), the other way around does not work, and it is not argued in much detail when this will fail.**
>
> We would like to remind the reviewer that the self-consistency condition (6) does imply (2) under fairly general conditions (if the velocity field is Lipschitz continuous and bounded) — this is discussed in Sec. 3.1.
>
> **Very few experiments were performed in the high-dimensional environment.**
>
> One challenge is that there is no closed-form ground truth solution for high-dimensional PDEs. We are only aware of OU processes which we tested extensively on. If the reviewer has suggestions on other high-dimensional PDEs that we could test our method on, please let us know.
>
> **What is the computational complexity of the method, with regard to the dimension of the PDEs?**
>
> Please see (c) in the common response to all reviewers.

---

> > ### Comment · Reviewer_egp4 · 2023-08-11
> >
> > Thanks for your response. Although OU process is one the very few that has a explicit solution, there are other PDEs with monte-carlo expressions, and can be evaluated relatively easily. For instance, the Black-Scholes equation has been used many places, like
> >
> > https://arxiv.org/abs/1804.07010
> >
> > I will leave my score as is.

---

> > > ### Author Response · Authors · 2023-08-14
> > >
> > > We thank the reviewer for pointing out the Black-Scholes equation (15) in the [referenced paper](https://arxiv.org/pdf/1804.07010.pdf). We would like to emphasize that the PDEs of our focus are mass-conserving, while the Black-Scholes equation is not: This can be seen from (16) that $\int u(t, x) dx$ is not a constant for varying t. Hence it is outside the scope of the present work.

---

### Official Review · Reviewer_wUYN · 2023-07-06

**Soundness:** 4 excellent
**Presentation:** 4 excellent
**Contribution:** 3 good
**Rating:** 6
**Confidence:** 4

**Summary:**

The paper proposes a method for solving mass-conserving partial differential equations (PDEs). The method leverages the property of self-consistency to devise an iterative optimization method for velocity matching. The proposed method was tested on several important PDEs.

**Strengths:**

- The paper is well-written. The motivation and the idea is clear.

- The description of the literature is comprehensive.

- Two sets of the proposed methods in very different characteristics, NODEs and TIPFs, are interesting.

- Empirical evidence is strong.


**Weaknesses:**

- As opposed to the very well-written texts, some of the figures are in poor quality. Figures 3, 4, 5 are particularly hard to read. It needs some work to make them more legible, e.g., different markers, etc.

- In-depth comparisons between SCVM-NODEs and SCVM-TIPFs would give more information/intuitions to readers on which method to use given a specific PDE problem. Please see Questions.

**Questions:**

As elaborated in the manuscript, NODEs and TIPFs have their own weaknesses: namely, training time for NODEs and scalability for TIPFs. Although some numbers, e.g., training time, are provided in some benchmark problems, it is hard to assess the pros and cons of the two approaches. Are there some systematic comparisons on this aspect? In-depth comparisons between SCVM-NODEs and SCVM-TIPFs?

**Limitations:**

The authors address the limitation to some extent, described in the form of future directions. One of those limitations is being asked in Questions.

---

> ### Author Rebuttal · Authors · 2023-08-08
>
> Thank you for your valuable review. Please find below our response to your specific questions.
>
> **Figures are of poor quality.**
>
> Thank you for pointing this out. We agree the figures are too small, partially due to the tight page limit. We will make the figures easier to read in the new revision.
>
> **In-depth comparison between NODE and TIPF would give more information/intuition.**
>
> Please see (c) in the common response to all reviewers.

---

> > ### Comment · Reviewer_wUYN · 2023-08-16
> >
> > Thank you for the authors' response.
> >
> > Could you provide any empirical evidences that support the pros/cons written in the global response, if there is any? Seeing some numbers obtained from experiments would give more information.
> > For example, the authors said "TIPF works well in low dimensions (e.g. OU process in Sec 4.2, Porous medium equation in Sec 4.3).". Could you clarify what do you mean by "TIPF works well"? In the paper, the authors provided information such as "ADJ takes 341 minutes to train, while SCVM-TIPF and 276 SCVM-NODE take 23 and 9 minutes respectively" in Section 4.2. These types of numbers for other experiments would allow readers to assess the performances of two different models.

---

> > > ### Author Response · Authors · 2023-08-16
> > >
> > > Thank you for the follow-up question. We would like to emphasize that we have quantitatively compared TIPF and NODE in Sec 4.2 and Sec 4.3 which support the claim that "TIPF works well in low dimensions".
> > >
> > > In the global response, by "OU process in Sec 4.2", we referred to Figure 3, the first two columns, where TIPF (blue) obtains lower symmetric KL and symmetric f-divergence than NODE (orange). To give you concrete numbers used in Figure 3,
> > > * In $d=5$, at $t=2.0$,
> > >     - TIPF attains $5.14 \times 10^{-5}$ and $5.43 \times 10^{-6}$ in symmetric KL and symmetric f-divergence respectively (averaged over 20 runs).
> > >     - NODE attains $3.81 \times 10^{-4}$ and $1.30 \times 10^{-4}$ in symmetric KL and symmetric f-divergence respectively.
> > > * In $d=10$, at $t=2.0$,
> > >     - TIPF attains $7.44 \times 10^{-4}$ and $6.58 \times 10^{-4}$ in symmetric KL and symmetric f-divergence respectively (averaged over 20 runs).
> > >     - NODE attains $7.48 \times 10^{-4}$ and $5.88 \times 10^{-4}$ in symmetric KL and symmetric f-divergence respectively.
> > >
> > > We see that in $d=5$, TIPF attains significantly lower metrics than NODE, but the numbers become comparable for both methods in $d=10$.
> > >
> > > In the global response, by "porous medium equation In Sec 4.3", we referred to Figure 4, right, where we see TIPF outperforms JKO methods. We do not use NODE here because it does not have easy access to $\nabla p^{m-2}$. This illustrates the advantage of TIPF which has exact density access compared to NODE.
> > >
> > > Most of the other pros and cons listed in the global response are inherent features/drawbacks of the two methods. For instance, NODE automatically satisfies the initial condition whereas TIPF needs an additional regularizer for this purpose.
> > >
> > > Please let us know if you have more questions.

---

> > > > ### Comment · Reviewer_wUYN · 2023-08-16
> > > >
> > > > Thank you for the response, again.
> > > >
> > > > - thanks for the clarification. I think the confusion mainly comes from the presentation. Again, the figures are really hard to interpret although the main point here is that, in $d=10$ (which is the high dim), there are no big advantages in using TIPF. For a better understanding/presentation of the results, it'd have been nicer to have a figure or table showing those measures (KL and f-div) over varying $d$ (not saying that the authors need to do it now in the middle of the discussion). Some systematic assessments of those two alternatives would make the paper stronger (e.g., performance in terms of those metrics + computational time for varying $d$).
> > > >
> > > > - although this is not a major point, the initial condition can be exactly enforced in TIPF as well (as shown in the work by Bilos, et al). Would there be some reasons why the authors do not consider such parameterization in TIPF (which allows to enforce the exact initial condition)?

---

> > > > > ### Author Response · Authors · 2023-08-17
> > > > >
> > > > > Thank you for the helpful comments.
> > > > >
> > > > > * We agree the figures are a bit hard to parse and we will definitely improve them in the next version. And having a figure or table showing varying $d$ is a good idea to demonstrate how TIPF falls off as $d$ increases --- we will incorporate that as well. Thanks for suggesting it.
> > > > >
> > > > > * You're right, and we have mentioned in passing that the initial condition can be enforced in TIPF as done by Eq.(6) in Bilos et al. Early on we experimented with this strategy but found that it works less well than using a regularizer, presumably because it makes the architecture too restrictive. We will mention this point more explicitly.

---

### Official Review · Reviewer_5yDs · 2023-07-07

**Soundness:** 2 fair
**Presentation:** 3 good
**Contribution:** 2 fair
**Rating:** 6
**Confidence:** 4

**Summary:**

The authors present a framework for solving mass-conserving partial differential equations (PDE) from Lagrangian viewpoint. In this framework the flow map or velocity field of the problem is parameterized by two neural network approaches: Time-dependent-Invertible-Push-Forward (TIPF) and Neural ODE (NODE). Then the self-consistency condition is enforced on the velocity field parameterized by the neural network methods mentioned above through an iterative optimization process. In this iterative procedure a biased estimate of the  gradient of the self-consistency condition is used. The proposed approach reduces the computational cost while providing high performance in terms of error metrics such as KL-divergence, total variation (TV) distance. The authors evaluate their framework by considering a wide range of PDEs and comparing against the baseline models developed in recent years.

**Strengths:**

- It is nice to see an extensive array of empirical evaluations against other baseline models.
- Overall the paper is well-written and easy to follow (despite few confusing statements addressed below)


**Weaknesses:**

- I wish there were some preliminary analysis on the effect of the biased gradient estimate. As authors clearly described in Appendix A, they eliminate some terms in the gradient of self-consistency loss (to alleviate some computational cost), and just use the term (13). To me, it seems like that term (14) can be computed in the same way as term (13) and I was wondering if there is any reason that it is eliminated. It is unclear how other terms could impact the model performance (accuracy) with the cost of increasing computational complexity of the model and how would this trade-off look like.
- I did not find any description of what boundary conditions were used for some of the problems such as Fokker-Planck equation and how much the model is capable of satisfying that. If they are not satisfied, how much is the effect on accuracy of the predicted solution.


**Questions:**

- In Section 3.2 it is mentioned that to enforce initial condition, an extra loss term in the form of L2 loss is included to the loss function for TIPF method. This does not guarantee the satisfaction of the constraint 100% and it is just a soft constraint enforcement that has been used by PINNs model as well. After the model is trained, did you check the model error for predicting the initial condition? Or after training process is finished, how does the initial condition loss trend looks like and how much is it at the end of training?
- In addition to initial condition, how do you handle other constraints such as boundary conditions? have you checked whether the predicted solution satisfies the boundary constraints, and if not, how much is the discrepancy?
- In page 4, I did not follow how the solution density $p_t$ can be recovered using the flow map. In particular, where did that equation come from: $log p_t(x) = log p_0^*(\phi_t^{-1}(x)) + log \text{det} | J \phi_t^{-1} (x)|$. It is important for authors to put themselves in readers' shoes and provide enough information when describing their methodology.
- Regarding the gradient estimate, could you provide some discussion about the missing terms in calculating the gradient? The proposed methods result in more accurate predictions compared to other baselines, does it imply that for these specific canonical problems the effect of those terms are negligible? I guess that may not be true. Is it useful to compare the predictions against accurate solutions from a numerical solver with fine time and space resolution for a stiff problem, in order to better capture the effect of missing terms?

**Limitations:**

- Authors describe a few future directions to investigate some existing limitations in the current manuscript. I would encourage them to first provide some numerical evidences about the potential limitations they want to explore in the future works. For example, as asked above, using their current framework, they can numerically investigate how much is the discrepancy in satisfying initial condition, and boundary conditions.
- Finding more accurate estimate of gradient in their iterative framework is another important aspect to be explored. For the current paper, I still hope to see that they provide some insight about the effect of missing terms.

---

> ### Author Rebuttal · Authors · 2023-08-08
>
> Thank you for your valuable review. Please find below our response to your specific questions.
>
> **To me, it seems like that term (14) can be computed in the same way as term (13) and I was wondering if there is any reason that it is eliminated.**
>
> Term (14) involves the term $J_\theta f_t^\theta(x) = J_\theta f_t(x; \mu_t^\theta)$. This term is challenging to compute because 1) the derivative of $f_t(x;\mu_t)$ w.r.t. $\mu_t$ can be complicated (e.g. in optimal transport formulation in Sec D.6), and 2) the derivative of $\mu_t^\theta$ w.r.t. $\theta$ can also be complicated: for instance, for NODE, numerical integration is needed to obtain $\mu_t^\theta$, so differentiating it w.r.t. $\theta$ must differentiate through each integration step which is costly; we verified this in our early experiments.
>
> In comparison, the term (13) is easy to compute because for both NODE and TIPF we have closed-form expressions for $v_t^\theta$.
>
> Please also refer to (a) in the common response to all reviewers.
>
> **Are boundary conditions used?**
>
> Aside from the initial condition, we did not consider any other boundary condition. This is in line with competing methods such as [Shen and Wang 2023, Mokrov et al. 2021, Fan et al. 2021]. As acknowledged in the conclusion, finding new ways to handle boundary conditions is an important future step.
>
> **After training SCVM-TIPF, what is the error in predicting the initial condition?**
>
> For OU process experiments, we have computed the same metrics from Figure 3 for the recovered flow at $t = 0$ — we did not include these metrics at $t=0$ because the plot is in log-scale and all other methods have exactly zero discrepancies at $t=0$. Since metrics are computed against the ground truth closed-form solution, the numbers at $t=0$ will reveal how well the initial condition is satisfied. For TIPF, the numbers for all metrics at $t=0$ are very small and are close to the numbers at $t=0.25$. For instance, for $d=10$, at $t=0$ the symmetric KL is $10^{-3.26}$ and the Bures-Wasserstein distance is $10^{-1.44}$. We have added a comment about this in Sec 4.2.
>
> For the porous media equation experiment, the first plot in Figure 4 shows visually the initial condition is well satisfied.
>
> **How is the density $p_t$ recovered from the flow map?**
>
> The formula $\log p_t(x) = \log p_0^*(\Phi_t^{-1}(x)) + \log \det |J\Phi_t^{-1}(x)|$ is a direct consequence of the change-of-variable formula. See (1) in [Dinh et al. 2016]. We have added this reference to the paper.
>
> **Further discussion about the missing terms in calculating the gradient?**
>
> Please refer to (a) in the common response to all reviewers.

---

### Official Review · Reviewer_v66q · 2023-07-07

**Soundness:** 4 excellent
**Presentation:** 3 good
**Contribution:** 3 good
**Rating:** 7
**Confidence:** 3

**Summary:**

This article deals with the problem of solving partial differential equations (PDEs) using neural networks. More specifically, the authors focus on mass-conserving PDEs:
$$ \forall x,\quad \forall t\in[0,T],\quad \partial_t p_t(x) = -\nabla\cdot( f_t(x;\mu_t)p_t ) . (Equation~2) $$

The authors have developed an iterative optimization scheme called "self-consistent velocity matching" (Algorithm 1 in the supplementary material). This method is based on self-consistency, _i.e._ by denoting $\Phi_t\colon\mathbb{R}^d\to\mathbb{R}^d$ the flow map at time $t$ associated to Equation (2) and $v_t\colon\mathbb{R}^d\to\mathbb{R}^d$ its velocity at time $t$, asking for
$$ \forall x,\quad v_t(x) = -f_t(x;\mu_t) . (Equation~6) $$

Assuming that the flow map $\Phi_t$ (or the velocity field $v_t$) is a neural network (TIPF for $\Phi_t$, or NODE for $v_t$) parametrized by a parameter $\theta$, the "self-consistent velocity matching" algorithm iterates three steps:
1. Define a space-time grid where to solve the equation: $\forall b=1,\ldots,B$, $\forall\ell=1,\ldots,L$, $~$ $x_b\sim\mu^\ast$, $t_\ell\sim\mathcal{U}([0,T])$,
2. Let particles $x_b$ drift along the flow $\Phi_t$: $\forall b$, $\forall\ell$, $~$ $y_{b,\ell}=\Phi_{t_\ell}^{\theta^k}(x_b)$,
3. Update the parameter $\theta^k$ according to a _biased_ gradient descent step: $\theta^{k+1}=\theta^k-\eta\frac1{BL} \nabla_\theta \sum_{b,\ell}\lVert v_{t_\ell}^{\theta}(y_{b,\ell})-f_{t_\ell}(y_{b,\ell};\mu^{\theta^k}_{t_\ell})\rVert^2$.

The authors conduct several numerical experiments to compare themselves with the state of the art. The results obtained are comparable or better in larger dimensions. In particular, the proposed method seems more in line with the laws of physics (_cf._ flow against obstacles experiment, Figure 13 in the supplementary material).

The theoretical convergence of the proposed method is not investigated.

**Strengths:**

- A self-sufficient article with precise references to the supplementary material that makes it easy to read.
- The proposed algorithm is precisely defined, step by step, and its limitations are discussed.
- In the supplementary material, all the details of the numerical implementation are given, making the experiments reproducible.


**Weaknesses:**

The introduction lacks context. Similarly, the supplementary material would benefit from being preceded by an opening paragraph with a title and summary of its contents.

The proposed method is based on the use of TIPF or NODE networks, each with its own advantages and disadvantages. It is not clear to me when to use one or the other.

**Questions:**

Some comments:
- Perhaps emphasize on lines 142-143 that the flow *or* the velocity field will be parameterized.
- Line 59: Which figure in Figure 2?
- Line 147: realNVP not defined
- The way references are written down is not consistent.
- When reference is made to figures in the supplementary material (Figure 8 and following), specify that they are in the supplementary material; the same applies to Algorithm 1.
- Algorithm 1: I think it should be $t_\ell$ instead of $t$ in the $\theta$ update (last line before *end for*).
- Equations (13), (14), and (15): Should there not be a 2?

**Limitations:**

Not concerned.

---

> ### Author Rebuttal · Authors · 2023-08-08
>
> Thank you for your valuable review. Please find below our response to your specific questions.
>
> **The introduction lacks context. Similarly, the supplementary material would benefit from being preceded by an opening paragraph with a title and summary of its contents.**
>
> Thank you for pointing this out. We have revised the text according to your comments and added an opening paragraph in the appendix as follows:
>
> “In this appendix, we provide details and further justification of the proposed method. In Appendix A, we provide an interpretation of the update (9) as a gradient descent step with a biased gradient. In Appendix B, we explain the integration-by-parts trick used to prove (3.1). In Appendix C, we provide implementation details of all considered methods. In Appendix D, we provide additional experimental details and results.”
>
> **The proposed method is based on the use of TIPF or NODE networks, each with its own advantages and disadvantages. It is not clear to me when to use one or the other.**
>
> Please refer to (c) in the common response to all reviewers.

---

### Official Review · Reviewer_3qHN · 2023-07-09

**Soundness:** 4 excellent
**Presentation:** 4 excellent
**Contribution:** 3 good
**Rating:** 7
**Confidence:** 3

**Summary:**

The authors propose a method for
solving mass-conserving partial differential
equations (PDEs) within an iterative optimisation scheme
 without resorting
to either spatial or temporal discretisation.

Inspired from previous work [1] on imposing
self-consistency for efficiently solving
Fokker-Planck equations, they
 impose a self-consistency constrain to
obtain the time-dependent velocity field that
is associated with mass conserving
PDE. Thus instead of solving a PDE in terms of a probability density, they solve more efficiently for obtaining the associated
velocity field.
They represent the probability flow of the mass-conserving PDE with a neural network, and within an iterative scheme they optimise
the weights of the network to satisfy the self-consistency constraint evaluating the relevant terms using the network weights of the previous iteration.

They test the performance of their approach
on several numerical experiments and compare
with analytic solutions (when available). The explored examples study diverse model systems starting from simple analytically tractable systems (Fokker-Planck equation with stationary density comprising a mixture of Gaussians and  Ornstein-Uhlenbeck processes of dimension $d=10;30;60$) , the porous medium and time-dependent Fokker-Planck equations, as well as some systems with many-body interacting dynamics, and smooth interpolation between non-trivial probability measures. Although the method of [1] seems to perform better in higher dimensnions, the current proposed approach claims to be considerably faster than [1].


**Strengths:**


- Impressive that the proposed approach does not require either time or space discretisation.

- well written and presented manuscript.

- applicable both for representing probability flows in terms of neural ODEs and as a time-dependent push forward, that making it possible to relate the current approach with several existing frameworks for solving PDEs/transforming prob. densities.

- the authors present extensive numerical experiments in multiple settings employing systems of relatively high dimension.

**Weaknesses:**


- Relatively small contribution when considering [1] and [2], but nevertheless impressive results.
- Some of the plots are extremely difficult to read [eg fig 4].

- There are no convergence guarantees of the framework and the solution relies on the convergence of the biased optimisation of the self consistency equation.

**Questions:**




- Do you have any insight what happens in Figure 2 in the d=30;60 experiments and the JKO-PD method has such a non monotonic decrease in sym KL with time?

- Could you probably perform a systematic study for the computational costs of the current approach for increasing system dimension and compare it with the other competing methods?

- Do you have any empirical insights on when the biased optimisation of the self-consistency constrain will fail?

Minor:

- There is a verb missing in Line 218
- Line 64 "comparing" seems incorrect here. Do you mean "competing"?

**Limitations:**

- No guarantee that the optimisation will converge on the correct solution given random weight initializations.

References:

[1] Shen, Zebang, and Zhenfu Wang. "Entropy-dissipation Informed Neural Network for McKean-Vlasov Type PDEs." arXiv preprint arXiv:2303.11205 (2023).

[2] Shen, Zebang, et al. "Self-consistency of the fokker planck equation." Conference on Learning Theory. PMLR, 2022.

---

> ### Author Rebuttal · Authors · 2023-08-08
>
> Thank you for your valuable review. Please find below our response to your specific questions.
>
> **Some of the plots are extremely difficult to read.**
>
> Thank you for pointing this out. We will make the figures bigger in the revision.
>
> **Although the method of [1] seems to perform better in higher dimensions, the current proposed approach claims to be considerably faster than [1].**
>
> We’d like to point out that the ADJ method of [1] *does not perform better* in higher dimensions (it becomes significantly worse than SCVM-NODE at $d=10$): this is shown in Figure 2 second row. As written in the first bullet point in Sec 4.1, we suspect this is due to the amount of error accumulated in the two integration passes in ADJ and the usage of up to third-order spatial derivatives of the neural velocity field. Although the formulation of ADJ has no bias, the numerical procedure can introduce more errors.
>
> **Do you have any insight into what happens in Figure 2 in the $d=30;60$ experiments and the JKO-PD method has such a non-monotonic decrease in sym KL with time?**
>
> JKO-PD is a primal-dual method that involves solving a min-max optimization problem. It is only equivalent to the JKO flow (for which the objective must be monotonic) if the inner maximization is solved perfectly. This is not the case in practice and is hence a disadvantage of JKO-PD. It is possible the result will be better with more inner steps. We use 2 outer minimization steps and 3 inner maximization steps, which is the default recommendation from the JKO-PD paper.
>
> **Could you probably perform a systematic study of the computational costs of the current approach for increasing system dimension and compare it with the other competing methods?**
>
> Please refer to (b) in the common response to all reviewers.
>
> **Do you have any empirical insights on when the biased optimization of the self-consistency constraint will fail?**
>
> For all experiments presented in the paper, the biased optimization is surprisingly effective without needing to tune the parameters. For SCVM-NODE, we did notice a failure case when the network parameters are initialized from a Gaussian with a very large variance (corresponding to a rapidly changing velocity field with high magnitude). However, the default weight initialization from FLAX suffices in all experiments. We also note other methods rely on proper weight initialization as well.
>
> Please also refer to (a) in the common response to all reviewers.

---

### Official Review · Reviewer_AfQf · 2023-07-23

**Soundness:** 4 excellent
**Presentation:** 4 excellent
**Contribution:** 3 good
**Rating:** 6
**Confidence:** 3

**Summary:**

This paper is concerned with solving mass-conserving PDEs using neural network parameterizations. The proposed methodology relies on the minimization self-consistency loss presented in [1, 2]. While the work of [2] is specialized to FP equation and McKean-Vlasov type PDEs, their approach cannot be generalized to more general cases. The author propose a simple yet effective way of minimizing the said loss. The proposed method is agnostic to the form of $f_t$ and furthermore, it does not require differentiating through $f_t$ and $\mu_\theta$, which is expensive in practice.

[1] Shen, Zebang, et al. "Self-consistency of the fokker planck equation." Conference on Learning Theory. PMLR, 2022.
[2] Shen, Z., & Wang, Z. (2023). Entropy-dissipation Informed Neural Network for McKean-Vlasov Type PDEs. arXiv preprint arXiv:2303.11205.



**Strengths:**

- I found the paper to be clearly written, well detailed and with an extensive discussion of the related works.
- The proposed method seems to have a significant advantage compared with JKO schemes and the adjoint method of [2]. First, it is simple to implement yet effective and fast. Next, for the FP equation or for the Wasserstein gradient flow, which are of practical interest, it is not required to differentiate through $\nabla \log p_t$ (which can be expensive).
- Overall, the numerical experiments are interesting and extensive.

**Weaknesses:**

I think that it may have been better to have a more detailed discussion/comment of the biased gradient estimate of the self-consistency loss $\nabla_\theta F(\theta, \theta_k)$. The resulting optimization procedure is reminiscent of a minimization-minimization procedure and I believe that there might be some connection that I haven't had time to dig in further. I guess that the paper would be a bit more interesting if the authors are able to justify that eq. (9) is a principled criterion that, when properly minimized, ensures the decrease of some statistical distance.

**Questions:**

l. 243, it is mentionned that TIPF, which involves normalizing flows, does not scale well with the dimension. This is surprising to me since normalizing flows do scale relatively well with the dimension. For $d = 60$ i do not see what the bottleneck is.

**Limitations:**

see above.

---

> ### Author Rebuttal · Authors · 2023-08-08
>
> Thank you for your valuable review. Please find below our response to your specific questions.
>
> **A detailed discussion of the biased gradient estimate and connection to the minimization-minimization procedure**
>
> Please refer to (a) in the common response to all reviewers.
>
> **Why doesn’t TIPF scale to $d=60$?**
>
> We use [RealNVP](https://arxiv.org/pdf/1605.08803.pdf) as the backbone for TIPF. The building blocks are coupling layers that update one coordinate while masking out the others. The bottleneck comes from the fact that we cannot afford too many coupling layers, since the number is proportional to the dimension. We have tested the option of updating multiple coordinates in one coupling layer but got significantly inferior results.
>
> In the original paper, RealNVP has been applied successfully to image space which is high-dimensional. For image space, it is possible to update many coordinates in a structured way (e.g. using a checkerboard pattern; see Figure 3 of the RealNVP paper) and still achieve good performance. However, we are dealing with unstructured vector data so similar tricks cannot be applied easily.
>
> It is possible that there exists a better choice of normalizing flow other than RealNVP, and TIPF could be improved with a better architecture. If you happen to know such an architecture, please let us know.

---

> > ### Comment · Reviewer_AfQf · 2023-08-15
> >
> > Thank you for your rebuttal. Please note that by "minimization-minimization" approach i meant an "Expectation-maximization" like approach. Other than this, I have no further question.

---

> > > ### Author Response · Authors · 2023-08-16
> > >
> > > Thank you for the clarification on "minimization-minimization". We want to point out that expectation-maximization can be viewed as coordinate descent which we discussed in the rebuttal point (a) --- see section "As a maximization–maximization procedure" on the [Wikipedia page](https://en.wikipedia.org/wiki/Expectation%E2%80%93maximization_algorithm).

---

### Author Rebuttal · Authors · 2023-08-08

We thank all reviewers for their constructive feedback. We appreciate the recognition of the simplicity and empirical effectiveness of our method. Please find below our common response to the initial reviews.

**(a) A more detailed discussion of the biased gradient estimate and other interpretations.**

Aside from Appendix A, it remains future work to quantify precisely the amount of bias in the biased gradient. As noted in Appendix A, a central challenge is to relate the Jacobians $J_\theta v_t^\theta$ and $J_\theta f_t^\theta$ in (13) and (14) for a given $\theta$. This is difficult because the derivatives are with respect to the neural network parameters, and hence the analysis must involve the specific network architecture.  Moreover, the dependency of $f_t^\theta$ on $\theta$ is via integrating along a velocity field parameterized by $\theta$ (if using NODE), which can be convoluted and hard to compute/analyze.

We have attempted a few alternative interpretations of the biased-gradient descent but they don’t offer any new insight. Reviewer AfQf suggested “minimization-minimization”, but we cannot find any reference to that; we suppose it is similar to coordinate descent. However, our iterates do not fit the framework of coordinate descent. To illustrate this difficulty, suppose we want to minimize $F(\theta, \theta’)$ defined in (9) using coordinate descent over two (block) coordinates $(\theta, \theta’)$. Then we cannot avoid computing the gradient of $F$ with respect to $\theta’$, which is very costly as argued in the previous paragraph.

**(b) The computational complexity of all methods with respect to the dimension.**

As with evaluating any optimization method, comparing computational cost is only meaningful when we control different methods to attain the same accuracy since increasing the computational cost (e.g. more iterations) would increase the accuracy. Since most presented methods in this work involve complicated training schemes and neural network parameterizations, it is hard to give a precise formula for the time complexity in terms of accuracy.

We can, however, comment on the time complexity of training for $S$ iterations for methods considered, in terms of dimension $d$. Assuming for simplicity that the same batch size is used, the training is done for $T$ time steps, and any network evaluation takes O(d) time,

- SCVM-NODE takes $O(ST N_{ODE}  d)$ time, where $N_{ODE}$ is the number of ODE integration steps.
- SCVM-TIPF (with RealNVP) takes $O(ST d^2)$ time since there will be $d$ coupling layers.
- ADJ takes $O(ST N_{ODE} d^3)$ time, where $d^3$ comes from the third-order spatial derivatives.
- JKO-ICNN takes $O(ST^2 d^3)$ time where $d^3$ is due to the log-determinant term.
- JKO-ICNN-PD takes $O(ST^2 d)$ time.

For JKO methods, the quadratic dependency on $T$ is due to the fact that, to draw samples from the current step, initial samples from the reference measure must be passed through all previous time steps. We also highlight that the inference time for JKO methods is significantly higher because a convex optimization needs to be solved to invert the ICNN flow for a total of $T$ times.

We will add a table about the time complexity of training for each method in the revised paper.


(c) A more in-depth comparison between SCVM-NODE and SCVM-TIPF.

Below is a summary of the advantages and disadvantages of TIPF and NODE. We will include this comparison in a new section in the appendix for the revised paper.

TIPS:
* Advantages:
  - TIPF works well in low dimensions (e.g. OU process in Sec 4.2, Porous medium equation in Sec 4.3).
  - It has exact and cheap density access, whereas NODE requires numerical integration to obtain the density.
* Disadvantages:
  - It does not scale up to high dimensions due to requiring many coupling layers, due to using RealNVP architecture.
  - The invertibility requirement of TIPF can limit its expressivity.
  - We need a regularizer to enforce the initial condition.

It is an interesting future direction to find alternatives to RealNVP that work better in higher dimensions.

NODE:
* Advantages:
  - NODE works well in high dimensions.
  - The requirement for the network parameterization is minimal (just needs to be a continuous vector field).
  - The Initial condition is automatically satisfied.
* Disadvantages:
  - Sampling and density queries require numerical integration, which can be slow and can incur numerical errors.

Our goal is to present two very different ways of parameterizing the probability flow to demonstrate the flexibility of the proposed generic framework (8-9). Finding new effective ways to parameterize the probability flow is an active research area, so we expect there will be better parameterizations in the future that can be plugged into our framework.

---

### Decision · Program_Chairs · 2023-09-21

**Decision:**

Accept (poster)

**Comment:**

The authors present a new approach to solving mass-conserving partial differential equations (PDEs) designing an iterative optimization scheme which avoids the need for spatial or temporal discretization. The main idea revolves around considering self-consistency constraints satisfied by the time-dependent velocity field associated with mass-conserving PDEs. This method then consists in learning an approximate velocity field based on these constraints, rather than directly solving the PDE in terms of a probability density.
The proposed approach is tested through a range of numerical experiments and is compared against analytic solutions whenever feasible.

The reviewers and I concur that the paper is an interesting contributions and should be accepted.

Please take into consideration reviewers' comments before submitting the final version of your work.